# TabStruct: Measuring Structural Fidelity of Tabular Data

## Abstract

Evaluating tabular generators remains a challenging problem, as the unique causal structural prior of heterogeneous tabular data does not lend itself to intuitive human inspection. Recent work has introduced structural fidelity as a tabular-specific evaluation dimension to assess whether synthetic data complies with the causal structures of real data. However, existing benchmarks often neglect the interplay between structural fidelity and conventional evaluation dimensions, thus failing to provide a holistic understanding of model performance. Moreover, they are typically limited to toy datasets, as quantifying existing structural fidelity metrics requires access to ground-truth causal structures, which is rarely available for real-world datasets. In this paper, we propose a novel evaluation framework that jointly considers structural fidelity and conventional evaluation dimensions. We introduce a new evaluation metric, *global utility*, which enables the assessment of structural fidelity even in the absence of ground-truth causal structures. In addition, we present *TabStruct*, a comprehensive evaluation benchmark offering large-scale quantitative analysis on 13 tabular generators from nine distinct categories, across 29 datasets. Our results demonstrate that global utility provides a task-independent, domain-agnostic lens for tabular generator performance. We release the TabStruct benchmark suite, including all datasets, evaluation pipelines, and raw results.

## 1 Introduction

Tabular data generation is a cornerstone of many real-world machine learning tasks [10, 29], ranging from training data augmentation [61, 23] to missing data imputation [98, 80]. These applications underscore the importance of generative modelling, which necessitates an appropriate understanding of the underlying data structure [50, 35, 9]. For instance, textual data conforms to the distributional hypothesis, and thus the autoregressive models are a natural workhorse for the text generation process [100, 77]. In contrast to the homogeneous modalities like text, tabular data can pose a different structural prior due to its heterogeneity – the features within a dataset typically have varying types and semantics, with feature sets that can differ across datasets [37, 80]. Recent work [41] on tabular foundation predictors has empirically demonstrated that the Structural Causal Model (SCM) is an effective structural prior of tabular data. As such, it is important to investigate how effectively existing generative models capture and leverage the causal structures of tabular data.

Prior work [40, 76, 25, 89, 59, 47] has attempted to assess tabular data generators by evaluating the synthetic data they produce. However, the prevailing evaluation paradigms still exhibit three primary limitations, which are summarised in Table 1: *(i) Insufficient tabular-specific fidelity assessments.* Current benchmarks largely adopt evaluation dimensions from homogeneous data modalities, such as density estimation [4], machine learning (ML) efficacy [94], and privacy preservation [53]. While effective in other modalities, they exhibit conceptual limitations when applied to tabular data – they do not explic-

Submitted to 39th Conference on Neural Information Processing Systems (NeurIPS 2025). Do not distribute.

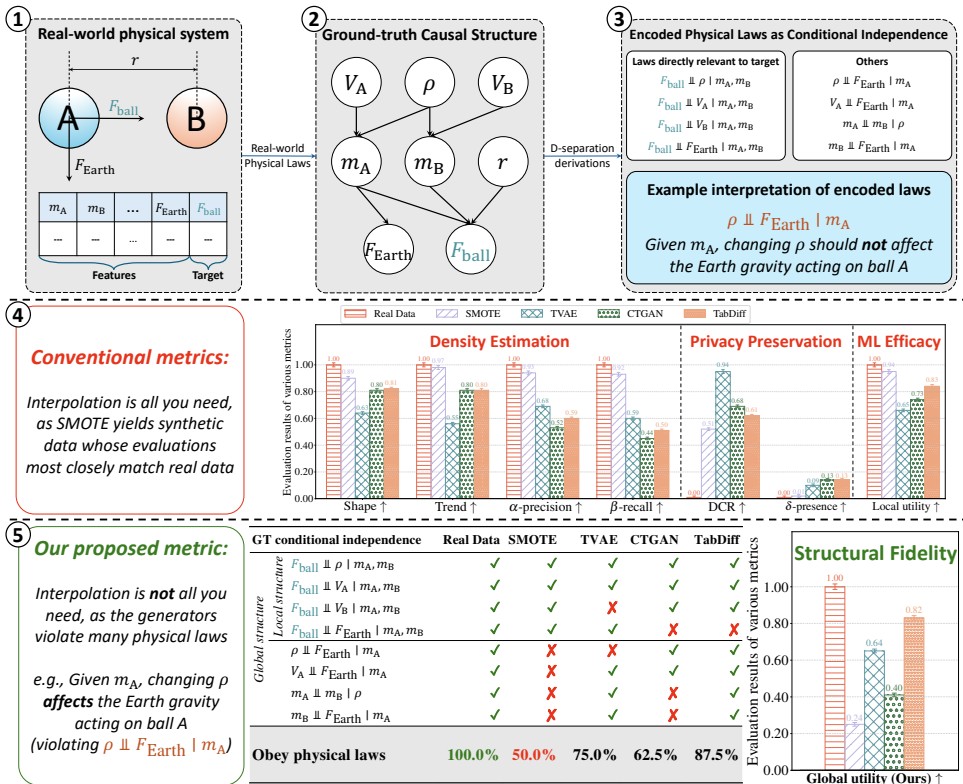

Figure 1: **Illustrative example highlighting the importance of fidelity check for tabular data structure.** ①: A real-world physical system showing the gravitational forces acting on ball A. The system is described by ball density ($\rho$), volume ($V$), masses ($m_A$ & $m_B$), distance ($r$), and gravitational forces ($F_{ball}$ & $F_{Earth}$). For simplicity, we assume both balls share identical density. ②: We derive the ground-truth (GT) causal structure of the system based on Newton's law of universal gravitation. ③: We interpret the encoded physical laws of the system as the conditional independence (CI) across variables. ④: We evaluate four generators by conventional metrics. ⑤: We assess the structural fidelity by CI tests and the proposed *global utility* metric. We note that the *global structure* reflects full conditional independence across all variables, while the *local structure* includes only those directly relevant to a specific prediction task at hand ($F_{ball}$). Results demonstrate that conventional metrics are insufficient: for instance, while SMOTE is able to outperform other generators on conventionally used dimensions (e.g., ML efficacy) – the generated synthetic data only preserves local structure and violates most physical laws. For tabular data, where the truthfulness and authenticity of synthetic data is hard to verify, global utility provides an effective mechanism for evaluating the alignment of the synthetic data to the likely ground-truth causal structure.

itly assess the unique structural prior of tabular data. A notable example is that many generators (e.g., SMOTE) can produce synthetic data with similar density estimation as real data, yet still violate underlying causal structures – such as physical laws illustrated in Figure 1(③). Although CauTabBench [89] takes a step forward to assess the structural fidelity of synthetic data, it remains confined to toy SCM datasets (i.e., synthetic datasets derived from random SCMs). Thus, CauTabBench offers limited insight into generative modelling performance on real-world tabular data, where the ground-truth SCMs are unavailable. *(ii) Potential evaluation biases.* Many benchmarks [40, 76] and model studies [94, 61, 98] prioritise ML efficacy as the principal dimension for assessing generator performance. For instance, in a classification setting, a generator is often considered effective if its synthetic data allows downstream models to achieve high predictive accuracy. However, while useful, ML efficacy can be highly sensitive to the choice of prediction task and target (Section 3.2.1). The reliance on ML efficacy can lead to biased conclusions: it tends to favour generators that are well-fitted for a specific prediction target, while obscuring their capacity to capture the global data structure (Figure 1(⑤)). *(iii) Limited evaluation scope.* Existing benchmarks mainly consider only a narrow range of datasets and generative models (Table 1), which restricts their ability to provide a thorough and generalisable comparison of model performance across the broader landscape of tabular generative modelling.

Table 1: **Evaluation scope comparison between TabStruct and prior tabular generative modelling benchmarks.** TabStruct presents a comprehensive evaluation framework for tabular generative models, incorporating a wide range of evaluation dimensions, datasets, and generator categories.

| Benchmark | Conventional dimensions | | | Structural fidelity | | Data | | Generator | |
|---|---|---|---|---|---|---|---|---|---|
| | Density Estimation | Privacy Preservation | ML Efficacy | SCM data | Real-world data | # Datasets | Contamination-free | # Models | # Categories |
| Hansen et al. [40] | ✔ | ✘ | ✔ | ✘ | ✘ | 11 | ✔ | 5 | 5 |
| Synthcity [76] | ✔ | ✔ | ✔ | ✘ | ✘ | 18 | ✘ | 6 | 4 |
| SynMeter [25] | ✔ | ✔ | ✔ | ✘ | ✘ | 12 | ✘ | 8 | 4 |
| CauTabBench [89] | ✔ | ✘ | ✘ | ✔ | ✘ | 10 | ✔ | 7 | 4 |
| SynthEval [59] | ✘ | ✔ | ✔ | ✘ | ✘ | 1 | ✔ | 5 | 3 |
| Karpar et al.[47] | ✔ | ✘ | ✔ | ✘ | ✘ | 2 | ✔ | 6 | 4 |
| **TabStruct (Ours)** | ✔ | ✔ | ✔ | ✔ | ✔ | **29** | ✔ | **13** | **9** |

In this paper, we aim to bridge these gaps by introducing a systematic and comprehensive evaluation framework for existing tabular generative models, with a particular focus on the structural prior of tabular data. Our proposed framework is characterised by five key concepts: (i) We explicitly incorporate **structural fidelity** of synthetic data as a core evaluation dimension for tabular generative models. Structural fidelity can directly reflect model capability in learning the structure of tabular data, without biasing towards a specific prediction target. In addition, we retain the three conventional evaluation dimensions (density estimation, privacy preservation, and ML efficacy) and investigate their interplay with structural fidelity, offering customised guidance for selecting suitable generators across diverse use cases. (ii) We evaluate structural fidelity on expert-validated SCM datasets. To ensure alignment with ground-truth causal structures, we avoid using toy SCMs and instead select SCM datasets with expert-validated causal structures. With ground-truth SCMs, we can derive the **conditional independence (CI)** of features. We then quantify structural fidelity through the difference in CI between real and synthetic data as shown in Figure 1(⑤) (iii) We further extend the evaluation of structural fidelity to real-world datasets, where the ground-truth SCMs are unavailable. To this end, we propose a novel evaluation metric, **global utility**, which treats each variable as a prediction target and measures how well it can be predicted using other variables. Importantly, global utility does not require ground-truth causal structures, thus enabling the evaluation of structural fidelity in real-world scenarios. (iv) We conduct an extensive empirical study on the performance of **13 tabular generators spanning nine categories on 29 datasets**, resulting in a total of **over 150,000 evaluations**. The large evaluation scope can ensure holistic and robust benchmarking results. (v) We introduce **TabStruct**, the benchmark suite developed for this work. TabStruct features a well-structured system design and consistent APIs for building and evaluating various tabular generative models. This open-source library aims to help the research community explore tabular generative modelling within a standardised framework.

Across both SCM and real-world datasets, our primary finding is:

*Structural fidelity, as quantified by the proposed global utility, should be a core dimension when evaluating tabular generative models.*

The benchmark results suggest the prevailing paradigm (i.e., optimising tabular generators primarily for improved density estimation and ML efficacy) is insufficient. In contrast, global utility offers a complementary perspective – tabular-specific fidelity assessments. Finally, we find that diffusion-based generators can be considered as a reliable approach for tabular data generation, given their consistent performance in capturing high-fidelity global data structures.

Our contributions can be summarised as follows:

- **Conceptual** (Section 3): We propose a unified evaluation framework for tabular generators that integrates structural fidelity with conventional dimensions, and introduce *global utility*, a novel metric that measures structural fidelity without requiring access to ground-truth causal structures.

- **Technical** (Section 3): We release the *TabStruct* benchmark suite[1], including datasets, generator implementations, evaluation pipelines, and all raw results.

- **Empirical** (Section 4): We conduct a large-scale quantitative study of 13 tabular generators on 29 datasets. The results offer actionable insights into model performance and can inspire the design of more effective tabular generators by attending to the unique structural prior of tabular data.

---

[1]Code is available at `https://anonymous.4open.science/r/TabStruct-E4E4`.

## 2 Related Work

**Tabular Generator Benchmarks.** An extensive line of benchmarks [86, 40, 76, 25, 49, 82, 60] has been proposed for tabular data generation, conventionally established around three dimensions: *density estimation*, *privacy preservation*, and *ML efficacy*. Density estimation [40, 4, 80, 98] assesses the divergence between real and synthetic data distributions. However, it fails to capture inter-feature interactions and, as a result, cannot evaluate whether synthetic data preserves the causal structures present in the real data. ML efficacy [94, 76, 79, 87] evaluates the performance difference when real data is replaced with synthetic data in downstream tasks, which primarily focuses on $p(y \mid \mathbf{x})$, thus inherently prioritising feature-target relationships over inter-feature interactions. Privacy preservation [25, 53, 42, 28], although essential in privacy-sensitive scenarios, is generally task-specific and usually does not necessitate high structural fidelity [20, 59, 67]. Recent efforts such as Synthcity [76] and SynMeter [25] have aimed to standardise the evaluation of tabular data generators by incorporating the three conventional dimensions. Nonetheless, they omit explicit assessment of tabular data structure. To the best of our knowledge, CauTabBench [89] is the only other benchmark to explicitly evaluate structural fidelity, but it is limited to toy SCM datasets, as existing metrics [16, 84] typically assume access to the ground-truth SCMs – a condition that is seldom satisfied and arguably infeasible for most real-world datasets [46, 34, 102]. We further provide a detailed summary of prior studies on tabular data generation in Appendix B. As shown in Table 1, despite the ongoing progress, existing benchmarks neither comprehensively cover all evaluation dimensions nor provide a broad evaluation scope across datasets and generators. To bridge these gaps, we introduce *global utility*, an SCM-free metric that quantifies how well a generator preserves the causal structure of real data. Our *TabStruct* benchmark provides a comprehensive evaluation framework for tabular generators.

## 3 Methods

### 3.1 Problem Setup

**Dataset and tabular generator.** Let $\mathcal{D}_{\text{full}} := \{(\mathbf{x}^{(i)}, y^{(i)})\}_{i=1}^{N} \sim p(\mathbf{x}, y)$ represent a labelled tabular dataset with $\mathbf{x}^{(i)} \in \mathbb{R}^D$. We refer to the $d$-th feature (i.e., a column/variable) as $\boldsymbol{x}_d$, and the $d$-th feature of the $i$-th sample (i.e., a cell) as $x_d^{(i)}$. For notational simplicity, we define $\boldsymbol{x}_{D+1} := \{y^{(i)}\}_{i=1}^{N}$, so that the full collection of variables, including both features and target, can be written as $\mathcal{X} := \{\boldsymbol{x}_1, \ldots, \boldsymbol{x}_D, \boldsymbol{x}_{D+1}\}$. We denote the training split of $\mathcal{D}_{\text{full}}$ as the reference dataset ($\mathcal{D}_{\text{ref}}$), and test data as $\mathcal{D}_{\text{test}}$. A tabular generator is trained on $\mathcal{D}_{\text{ref}}$ and aims to generate synthetic data $\mathcal{D}_{\text{syn}} \sim p(\tilde{\mathbf{x}}, \tilde{y})$ close to $p(\mathbf{x}, y)$. We evaluate the quality of $\mathcal{D}_{\text{ref}}$ wrt. all the metrics, thus providing a benchmark performance against which $\mathcal{D}_{\text{syn}}$ is compared. We refer to any dataset being assessed as "evaluation dataset $\mathcal{D}$", thus, both $\mathcal{D}_{\text{ref}}$ and $\mathcal{D}_{\text{syn}}$ may serve as evaluation datasets.

**Structural causal models (SCM).** Under the assumptions of causal sufficiency, the Markov property, and faithfulness, an SCM is defined by the quadruple $M := \langle \mathcal{X}, \mathcal{G}, \mathcal{F}, \mathcal{E} \rangle$. $\mathcal{G}$ is the causal graph that encodes the causal relationships among the variables. $\mathcal{E} := \{\boldsymbol{\epsilon}_j\}_{j=1}^{D+1}$ denotes the exogenous noise, and $\mathcal{F} := \{f_j\}_{j=1}^{D+1}$ is the set of structural functions. Each variable $\boldsymbol{x}_j$ is determined by a function $f_j$ of its parents and its exogenous noise, that is, $\boldsymbol{x}_j = f_j(\text{pa}(\boldsymbol{x}_j), \boldsymbol{\epsilon}_j)$, where $\text{pa}(\boldsymbol{x}_j) \subseteq \mathcal{X} \setminus \{\boldsymbol{x}_j\}$ denotes the parent set of $\boldsymbol{x}_j$ in the graph $\mathcal{G}$.

### 3.2 Structural Fidelity

As an empirically effective structural prior for tabular data, SCM provides a formal framework for the underlying generative processes of tabular data [41, 89]. Therefore, we define the *structural fidelity* of a tabular generator as the alignment between the SCMs in its synthetic data and the ground-truth causal structures. Next, we introduce the quantifications of structural fidelity on SCM (Section 3.2.1) and real-world (Section 3.2.2) datasets. We further discuss the rationales behind using causal structural prior for tabular data in Appendix D.

#### 3.2.1 Conditional Independence Score: Quantifying Structural Fidelity with SCM

**Motivation.** We begin by quantifying structural fidelity under the assumption that the ground-truth SCM is available. Following established benchmarks in causal discovery and inference [84, 46, 89], we evaluate structural fidelity at the level of the Markov equivalence class. At this level, causal structures are represented as completed partially directed acyclic graphs (CPDAGs). The SCMs of $\mathcal{D}_{\text{ref}}$ and $\mathcal{D}_{\text{syn}}$ are equivalent if they entail the same set of conditional independence (CI) statements (see Figure 1(② & ③) for an illustration). This implies that both SCMs serve as minimal I-MAPs [2] of the

joint distribution factorisation $p(\mathcal{X}) = \prod_{j=1}^{D+1} p(\boldsymbol{x}_j \mid \mathrm{pa}\,(\boldsymbol{x}_j))$, and no causal directions can be further removed. Therefore, the CPDAG-level evaluation provides a lens to interpret the fidelity of the tabular data. Further discussion on the rationale for CPDAG-level evaluation is provided in Appendix D.

**CI scores at various granularities.** Following prior work [84, 89], the full set of CI statements implied by the ground-truth SCM on $\mathcal{D}_{\mathrm{ref}}$ is defined as

$$\mathcal{C}_{\mathrm{global}} := \left\{ (\boldsymbol{x}_j \perp\!\!\!\perp \boldsymbol{x}_k \mid S_{j,k}) \mid S_{j,k} \subseteq \mathcal{X} \setminus \{\boldsymbol{x}_j, \boldsymbol{x}_k\} \right\} \cup \left\{ (\boldsymbol{x}_j \not\perp\!\!\!\perp \boldsymbol{x}_k \mid \hat{S}_{j,k}) \mid \hat{S}_{j,k} \subsetneq S_{j,k} \right\} \quad (1)$$

where $S_{j,k}$ and $\hat{S}_{j,k}$ are the d-separation and d-connection sets for $(\boldsymbol{x}_j, \boldsymbol{x}_k)$, respectively. The derivations of CI statements are fully programmatic [85, 24, 21]. More details are in Appendix C.

For each CI statement, we assess whether it holds in the evaluation dataset $\mathcal{D}$ (i.e., $\mathcal{D}_{\mathrm{ref}}$ or $\mathcal{D}_{\mathrm{syn}}$) by conducting a CI test at the significance level $\alpha = 0.01$ via

$$\widehat{\mathcal{I}}_\alpha(\boldsymbol{x}_j, \boldsymbol{x}_k \mid S_{j,k}, \hat{S}_{j,k}; \mathcal{D}) = \begin{cases} 1, & \text{if the CI statement is } not \text{ rejected on } \mathcal{D} \text{ at level } \alpha, \\ 0, & \text{otherwise.} \end{cases} \quad (2)$$

To quantify structural fidelity at varying levels of granularity, we define the CI score for any subset of CI statements $\mathcal{C} \subseteq \mathcal{C}_{\mathrm{global}}$ as:

$$\mathrm{CI}\,(\mathcal{C}, \mathcal{D}) \;=\; \frac{1}{|\mathcal{C}|} \sum_{\mathcal{C}} \mathbb{1}\left[ \widehat{\mathcal{I}}_\alpha(\boldsymbol{x}_j, \boldsymbol{x}_k \mid S_{j,k}, \hat{S}_{j,k}; \mathcal{D}) \;=\; 1 \right] \quad (3)$$

where $\mathrm{CI}\,(\mathcal{C}, \mathcal{D}) \in [0, 1]$ measures the fidelity of selected CI statements in $\mathcal{D}$, and $\mathbb{1}(\cdot)$ denotes the indicator function. A higher CI score indicates that the evaluation dataset more closely aligns with the structure of the ground-truth SCM. Implementation details for the CI scores are in Appendix C.

**Local structure vs. Global structure.** We assess structural fidelity at two levels of granularity: local and global. For local structural fidelity, we define the **local CI** score, $\mathrm{CI}\,(\mathcal{C}_{\mathrm{local}}, \mathcal{D})$, by considering only the CI statements that directly involve the prediction target $y$ of a given dataset and predictive task. Specifically, we compute the local CI score using Equation (3) with $\mathcal{C}_{\mathrm{local}} = \big\{ (\boldsymbol{x}_j \perp\!\!\!\perp \boldsymbol{x}_{D+1} \mid S_{j,D+1}) \mid j \in [D] \big\} \cup \big\{ (\boldsymbol{x}_j \not\perp\!\!\!\perp \boldsymbol{x}_{D+1} \mid \hat{S}_{j,D+1}) \mid j \in [D] \big\}$ (see Figure 1(③) for an illustration). $\mathcal{C}_{\mathrm{local}}$ highlights which features are uninformative for predicting $y$ when conditioned on the corresponding d-separation sets. Therefore, matching the local CI set indicates which features should be ignored when learning $p(y \mid \mathbf{x})$. A higher local CI score suggests the generator faithfully captures the local structure around the target, implying the potentially high utility of $\mathcal{D}$ for downstream predictive tasks. We empirically observe a strong correlation between the local CI score and the predictive performance on $y$ (Section 4.2).

For global structural fidelity, we define the **global CI** score as the CI score computed over the full set of CI statements, that is, $\mathrm{CI}\,(\mathcal{C}_{\mathrm{global}}, \mathcal{D})$. Global CI provides a comprehensive assessment of the entire causal structure encoded in the dataset, mitigating potential bias towards any particular variable.

### 3.2.2 Global Utility: SCM-free Metric for Global Structural Fidelity

**Motivation.** The CI scores introduced in Section 3.2.1 require access to a ground-truth SCM to enumerate the CI statements $\mathcal{C}_{\mathrm{global}}$. However, for real-world datasets, such an SCM is typically unavailable or even non-identifiable, thereby precluding direct evaluation of structural fidelity. To address this limitation, we propose *global utility* as an SCM-free proxy for global CI.

**Utility per variable.** Given an evaluation dataset $\mathcal{D}$, we treat each variable $\boldsymbol{x}_j \in \mathcal{X}$ as a prediction target. An ensemble of multiple downstream predictors is trained to predict $\boldsymbol{x}_j$ using the remaining variables $\mathcal{X} \setminus \{\boldsymbol{x}_j\}$ as inputs, following a standard supervised learning setup. The predictive performance on $\mathcal{D}_{\mathrm{test}}$ is denoted as $\mathrm{Perf}_j(\mathcal{D})$, measured using *balanced accuracy* for categorical variables and *root mean square error (RMSE)* for numerical variables. We define the utility of variable $\boldsymbol{x}_j$ as the relative performance achieved on evaluation data compared to reference data:

$$\mathrm{Utility}_j(\mathcal{D}) := \begin{cases} \mathrm{Perf}_j(\mathcal{D}_{\mathrm{ref}})^{-1} \mathrm{Perf}_j(\mathcal{D}), & \text{if } \boldsymbol{x}_j \text{ is categorical,} \\[2mm] \mathrm{Perf}_j(\mathcal{D})^{-1} \mathrm{Perf}_j(\mathcal{D}_{\mathrm{ref}}), & \text{if } \boldsymbol{x}_j \text{ is numerical.} \end{cases} \quad (4)$$

Utility offers a unified perspective for interpreting downstream performance across mixed variable types: $\mathrm{Utility}_j \geq 1$ indicates that downstream predictors trained on $\mathcal{D}$ perform on par with or better

than those trained on $\mathcal{D}_{\text{ref}}$ for predicting $\boldsymbol{x}_j$, whereas $\text{Utility}_j < 1$ implies a loss in predictive power. To mitigate the potential bias from a specific downstream predictor, we ensemble nine different predictors with AutoGluon [27]. Full technical details are in Appendix C.

**Global utility.** The theoretical (Section 3.2.1) and empirical (Section 4.2) analysis showcases a strong correlation between the local CI score ($\text{CI}\,(\mathcal{C}_{\text{local}}, \mathcal{D})$) and the predictive performance of $y$ ($\text{Utility}_{D+1}(\mathcal{D})$). Therefore, we hypothesise that aggregating the utility across all features can approximate the global CI score ($\text{CI}\,(\mathcal{C}_{\text{global}}, \mathcal{D})$), and we define the global utility as: $\text{Global Utility}(\mathcal{D}) \coloneqq \frac{1}{D+1} \sum_{j=1}^{D+1} \text{Utility}_j(\mathcal{D})$. Global utility is grounded in the observation that a high-fidelity generator should enable accurate conditional prediction of each variable from the others – an idea closely tied to the Markov blanket in SCMs [32, 33]. Our experiments reveal a strong correlation between global CI and global utility (Section 4.2), supporting that global utility serves as an effective and practical metric for evaluating global structural fidelity in the absence of ground-truth SCMs.

**ML efficacy and local utility.** The utility of the prediction target, $\text{Utility}_{D+1}(\mathcal{D})$, commonly referred to as local utility, aligns with the standard metric for assessing ML efficacy of tabular data generators. However, both theoretically (Section 3.2.1) and empirically (Section 4.2), we demonstrate that local utility can be biased, and even fail to reflect the model's ability to capture the full causal structure. In contrast, our proposed global utility mitigates this limitation by treating each feature fairly, thereby enabling a more robust and comprehensive evaluation of structural fidelity.

### 3.3 TabStruct Benchmark Suite

To address the limited evaluation scope of existing benchmarks, we propose **TabStruct**, a novel benchmark suite that jointly considers structural fidelity alongside conventional evaluation dimensions, and offers practical insights into real-world scenarios. Detailed descriptions are in Appendix F.

**SCM datasets.** To reduce the gap between causal structures in SCM and real-world data, we select six expert-validated SCM datasets from `bnlearn` [78], containing 7-64 features. Full dataset descriptions are provided in Appendix E.

**Real-world datasets.** We observe that many existing generators achieve near-perfect performance on commonly used benchmark datasets [80, 98], suggesting that these datasets offer limited discriminative power. To address this, we select 14 classification datasets from the hard TabZilla suite [26], containing 846-98,050 samples and 6-145 features. We further select nine challenging regression datasets, containing 345-22,784 samples and 6-82 features. Following prior work [61], we exclude any datasets employed for meta-validation of TabPFN to prevent data contamination, as TabPFN is used to compute utility scores. Full dataset descriptions are available in Appendix E.

**Benchmark generators.** TabStruct includes 13 existing tabular data generation methods of nine different categories: (i) a standard interpolation method SMOTE [15]; (ii) a structure learning method Bayesian Network (BN) [76]; (iii) two Variational Autoencoders (VAE) based methods TVAE [94] and GOGGLE [58]; (iv) a Generative Adversarial Networks (GAN) method CTGAN [94]; (v) a normalising flow model Neural Spine Flows (NFLOW) [26]; (vi) a tree-based method Adversarial Random Forests (ARF) [92]; (vii) three diffusion models: TabDDPM [53], TabSyn [98], TabDiff [80]; (viii) two energy-based models: TabEBM [61] and NRGBoost [12]; and (ix) a Large Language Model (LLM) based method GReaT [11]. In addition, we include $\mathcal{D}_{\text{ref}}$, where the reference data is used directly for evaluation. Full implementation details of benchmark generators are in Appendix E.

## 4 Experiments

We evaluate 13 tabular generators on 29 datasets by focusing on four research questions, and we further provide promising directions and practical guidance for developing tabular generative models across various use cases in Appendix G.

- **Validity of Benchmark Framework (Q1)**: Can the proposed evaluation framework, including the selected datasets and metrics, yield valid evaluation results regarding generator performance?
- **Validity of Global Utility (Q2)**: Can global utility serve as an effective metric for structural fidelity when ground-truth causal structures are unavailable?
- **Structural Fidelity of Generators (Q3)**: Can existing tabular generators accurately capture the underlying data structures across both SCM and real-world datasets?
- **Practicability of Global Utility (Q4)**: Can global utility provide stable and computationally feasible evaluation results for structural fidelity?

Table 2: **Benchmark results of 13 tabular generators on 29 datasets.** We report the normalised mean ± std metric values across datasets. "N/A" denotes that a specific metric is not applicable. We highlight the **First**, **Second** and **Third** best performances for each metric. For visualisation, we abbreviate "conditional independence" as "CI". The results show that the Top-3 methods in *Global CI* and *Global utility* are largely consistent between SCM and real-world datasets. This alignment suggests that the selected SCM datasets represent real-world causal structure, and global utility can serve as an effective proxy for global CI to evaluate global structural fidelity.

| Generator | Density Estimation | | | | Privacy Preservation | | ML Efficacy | Structural Fidelity | | |
|---|---|---|---|---|---|---|---|---|---|---|
| | Shape ↑ | Trend ↑ | $\alpha$-precision ↑ | $\beta$-recall ↑ | DCR ↑ | $\delta$-Presence ↑ | Local utility ↑ | Local CI ↑ | Global CI ↑ | Global utility ↑ |
| **SCM datasets** | | | | | | | | | | |
| $\mathcal{D}_{ref}$ | $1.00_{\pm0.00}$ | $1.00_{\pm0.00}$ | $1.00_{\pm0.00}$ | $1.00_{\pm0.00}$ | $0.00_{\pm0.00}$ | $0.00_{\pm0.00}$ | $0.99_{\pm0.01}$ | $0.89_{\pm0.10}$ | $1.00_{\pm0.00}$ | $0.99_{\pm0.01}$ |
| SMOTE | $0.82_{\pm0.09}$ | $0.85_{\pm0.06}$ | $0.60_{\pm0.17}$ | $0.83_{\pm0.01}$ | $0.21_{\pm0.09}$ | $0.01_{\pm0.01}$ | $0.92_{\pm0.07}$ | $0.82_{\pm0.12}$ | $0.30_{\pm0.11}$ | $0.39_{\pm0.09}$ |
| BN | $0.80_{\pm0.09}$ | $0.73_{\pm0.10}$ | $0.78_{\pm0.10}$ | $0.32_{\pm0.08}$ | $0.65_{\pm0.16}$ | $0.07_{\pm0.05}$ | $0.41_{\pm0.17}$ | $0.23_{\pm0.12}$ | $0.35_{\pm0.20}$ | $0.49_{\pm0.24}$ |
| TVAE | $0.59_{\pm0.10}$ | $0.59_{\pm0.14}$ | $0.65_{\pm0.14}$ | $0.36_{\pm0.06}$ | $0.70_{\pm0.10}$ | $0.13_{\pm0.11}$ | $0.78_{\pm0.14}$ | $0.50_{\pm0.21}$ | $0.40_{\pm0.09}$ | $0.70_{\pm0.11}$ |
| GOGGLE | $0.46_{\pm0.16}$ | $0.50_{\pm0.13}$ | $0.47_{\pm0.20}$ | $0.36_{\pm0.09}$ | $0.55_{\pm0.13}$ | $0.38_{\pm0.19}$ | $0.53_{\pm0.06}$ | $0.42_{\pm0.27}$ | $0.14_{\pm0.03}$ | $0.24_{\pm0.08}$ |
| CTGAN | $0.46_{\pm0.14}$ | $0.50_{\pm0.16}$ | $0.71_{\pm0.13}$ | $0.34_{\pm0.08}$ | $0.52_{\pm0.11}$ | $0.19_{\pm0.15}$ | $0.80_{\pm0.11}$ | $0.61_{\pm0.08}$ | $0.08_{\pm0.04}$ | $0.26_{\pm0.10}$ |
| NFlow | $0.31_{\pm0.15}$ | $0.26_{\pm0.10}$ | $0.31_{\pm0.21}$ | $0.15_{\pm0.09}$ | $0.73_{\pm0.16}$ | $0.51_{\pm0.13}$ | $0.10_{\pm0.05}$ | $0.09_{\pm0.07}$ | $0.09_{\pm0.07}$ | $0.12_{\pm0.07}$ |
| ARF | $0.75_{\pm0.14}$ | $0.71_{\pm0.11}$ | $0.79_{\pm0.09}$ | $0.36_{\pm0.09}$ | $0.50_{\pm0.13}$ | $0.09_{\pm0.07}$ | $0.57_{\pm0.04}$ | $0.21_{\pm0.09}$ | $0.35_{\pm0.11}$ | $0.68_{\pm0.11}$ |
| TabDDPM | $0.62_{\pm0.11}$ | $0.60_{\pm0.12}$ | $0.64_{\pm0.19}$ | $0.39_{\pm0.09}$ | $0.44_{\pm0.19}$ | $0.14_{\pm0.05}$ | $0.29_{\pm0.06}$ | $0.17_{\pm0.08}$ | $0.69_{\pm0.08}$ | $0.80_{\pm0.05}$ |
| TabSyn | $0.50_{\pm0.16}$ | $0.48_{\pm0.17}$ | $0.59_{\pm0.14}$ | $0.31_{\pm0.11}$ | $0.45_{\pm0.14}$ | $0.32_{\pm0.21}$ | $0.76_{\pm0.05}$ | $0.70_{\pm0.06}$ | $0.70_{\pm0.04}$ | $0.76_{\pm0.06}$ |
| TabDiff | $0.69_{\pm0.11}$ | $0.62_{\pm0.15}$ | $0.75_{\pm0.09}$ | $0.36_{\pm0.09}$ | $0.50_{\pm0.14}$ | $0.13_{\pm0.03}$ | $0.80_{\pm0.06}$ | $0.58_{\pm0.14}$ | $0.57_{\pm0.15}$ | $0.75_{\pm0.07}$ |
| TabEBM | $0.67_{\pm0.12}$ | $0.57_{\pm0.15}$ | $0.76_{\pm0.04}$ | $0.27_{\pm0.09}$ | $0.55_{\pm0.22}$ | $0.14_{\pm0.06}$ | $0.59_{\pm0.05}$ | $0.50_{\pm0.19}$ | $0.26_{\pm0.11}$ | $0.30_{\pm0.08}$ |
| NRGBoost | $0.65_{\pm0.10}$ | $0.50_{\pm0.15}$ | $0.61_{\pm0.14}$ | $0.26_{\pm0.07}$ | $0.53_{\pm0.12}$ | $0.28_{\pm0.21}$ | $0.75_{\pm0.01}$ | $0.64_{\pm0.05}$ | $0.11_{\pm0.05}$ | $0.16_{\pm0.02}$ |
| GReaT | $0.62_{\pm0.09}$ | $0.59_{\pm0.07}$ | $0.62_{\pm0.10}$ | $0.38_{\pm0.07}$ | $0.52_{\pm0.07}$ | $0.18_{\pm0.05}$ | $0.27_{\pm0.09}$ | $0.17_{\pm0.04}$ | $0.16_{\pm0.05}$ | $0.25_{\pm0.08}$ |
| **Real-world datasets** | | | | | | | | | | |
| $\mathcal{D}_{ref}$ | $1.00_{\pm0.00}$ | $1.00_{\pm0.00}$ | $1.00_{\pm0.00}$ | $1.00_{\pm0.00}$ | $0.00_{\pm0.00}$ | $0.00_{\pm0.00}$ | $0.96_{\pm0.06}$ | N/A | N/A | $0.99_{\pm0.01}$ |
| SMOTE | $0.61_{\pm0.13}$ | $0.87_{\pm0.05}$ | $0.81_{\pm0.11}$ | $0.77_{\pm0.01}$ | $0.19_{\pm0.04}$ | $0.02_{\pm0.02}$ | $0.91_{\pm0.07}$ | N/A | N/A | $0.41_{\pm0.04}$ |
| BN | $0.66_{\pm0.11}$ | $0.72_{\pm0.09}$ | $0.86_{\pm0.09}$ | $0.30_{\pm0.04}$ | $0.48_{\pm0.16}$ | $0.07_{\pm0.08}$ | $0.38_{\pm0.16}$ | N/A | N/A | $0.44_{\pm0.25}$ |
| TVAE | $0.45_{\pm0.20}$ | $0.50_{\pm0.14}$ | $0.55_{\pm0.20}$ | $0.18_{\pm0.04}$ | $0.68_{\pm0.18}$ | $0.29_{\pm0.18}$ | $0.70_{\pm0.06}$ | N/A | N/A | $0.53_{\pm0.13}$ |
| GOGGLE | $0.41_{\pm0.15}$ | $0.47_{\pm0.14}$ | $0.57_{\pm0.16}$ | $0.26_{\pm0.07}$ | $0.50_{\pm0.18}$ | $0.35_{\pm0.18}$ | $0.46_{\pm0.04}$ | N/A | N/A | $0.21_{\pm0.06}$ |
| CTGAN | $0.29_{\pm0.18}$ | $0.53_{\pm0.14}$ | $0.66_{\pm0.21}$ | $0.11_{\pm0.05}$ | $0.51_{\pm0.13}$ | $0.30_{\pm0.24}$ | $0.70_{\pm0.06}$ | N/A | N/A | $0.13_{\pm0.06}$ |
| NFlow | $0.38_{\pm0.19}$ | $0.28_{\pm0.16}$ | $0.52_{\pm0.15}$ | $0.07_{\pm0.04}$ | $0.64_{\pm0.14}$ | $0.42_{\pm0.25}$ | $0.10_{\pm0.06}$ | N/A | N/A | $0.14_{\pm0.12}$ |
| ARF | $0.61_{\pm0.11}$ | $0.58_{\pm0.12}$ | $0.83_{\pm0.10}$ | $0.21_{\pm0.04}$ | $0.48_{\pm0.14}$ | $0.05_{\pm0.04}$ | $0.54_{\pm0.07}$ | N/A | N/A | $0.56_{\pm0.12}$ |
| TabDDPM | $0.43_{\pm0.16}$ | $0.49_{\pm0.18}$ | $0.54_{\pm0.22}$ | $0.26_{\pm0.09}$ | $0.42_{\pm0.19}$ | $0.27_{\pm0.18}$ | $0.27_{\pm0.06}$ | N/A | N/A | $0.72_{\pm0.08}$ |
| TabSyn | $0.44_{\pm0.14}$ | $0.51_{\pm0.16}$ | $0.62_{\pm0.18}$ | $0.24_{\pm0.08}$ | $0.51_{\pm0.12}$ | $0.24_{\pm0.14}$ | $0.76_{\pm0.08}$ | N/A | N/A | $0.73_{\pm0.07}$ |
| TabDiff | $0.54_{\pm0.15}$ | $0.52_{\pm0.16}$ | $0.69_{\pm0.12}$ | $0.22_{\pm0.07}$ | $0.57_{\pm0.15}$ | $0.20_{\pm0.13}$ | $0.78_{\pm0.03}$ | N/A | N/A | $0.73_{\pm0.07}$ |
| TabEBM | $0.59_{\pm0.15}$ | $0.65_{\pm0.08}$ | $0.79_{\pm0.04}$ | $0.30_{\pm0.10}$ | $0.58_{\pm0.16}$ | $0.14_{\pm0.03}$ | $0.63_{\pm0.11}$ | N/A | N/A | $0.35_{\pm0.11}$ |
| NRGBoost | $0.54_{\pm0.12}$ | $0.49_{\pm0.13}$ | $0.62_{\pm0.16}$ | $0.20_{\pm0.07}$ | $0.51_{\pm0.15}$ | $0.22_{\pm0.13}$ | $0.74_{\pm0.05}$ | N/A | N/A | $0.16_{\pm0.05}$ |
| GReaT | $0.47_{\pm0.10}$ | $0.49_{\pm0.13}$ | $0.57_{\pm0.14}$ | $0.26_{\pm0.08}$ | $0.52_{\pm0.11}$ | $0.27_{\pm0.15}$ | $0.23_{\pm0.07}$ | N/A | N/A | $0.20_{\pm0.06}$ |

**Experimental setup.** For each dataset of $N$ samples, we first split it into train and test sets (80% train and 20% test). We further split the train set into a training split ($\mathcal{D}_{ref}$) and a validation split (90% training and 10% validation). We repeat the splitting 10 times, summing up to 10 runs per dataset. All benchmark generators are trained on $\mathcal{D}_{ref}$, and each generator produces a synthetic dataset with $N_{ref}$ samples. We tune the parameterised generators using Optuna [3] to minimise their average validation loss across 10 repeated runs. Each generator is given at most two hours to complete a single repeat. The reported results are averaged by default over 10 repeats. We aggregate results across all SCM or real-world datasets because the findings are consistent across classification and regression tasks. Specifically, we use the average distance to the minimum (ADTM) metric via affine renormalisation between the top-performing and worse-performing models [37, 63, 41, 61, 44]. We further provide the detailed configurations (Appendix E) and raw results (Appendix H).

## 4.1 Validity of Benchmark Framework (Q1)

**The benchmark metrics effectively evaluate data quality.** Table 2 demonstrates that all metrics effectively distinguish between high- and low-quality data. Specifically, except for privacy-related metrics, the reference data ($\mathcal{D}_{ref}$) consistently achieves the highest scores. This is expected, as $\mathcal{D}_{ref}$ is the ground truth and should score highly on metrics of density estimation, ML efficacy, and structural fidelity. In contrast, privacy metrics reward greater differences from the ground truth to indicate stronger privacy preservation. Since $\mathcal{D}_{ref}$ is identical to the ground truth, it naturally scores poorly for privacy. These results show that the selected metrics provide appropriate evaluations for data quality. Therefore, we consider the evaluation results to be valid and meaningful for analysis.

**The benchmark datasets present a genuine challenge for existing generators.** As detailed in Section 3.3, we select challenging, contamination-free real-world datasets, ensuring that they are non-trivial for existing tabular data generators. Table 2 illustrates that, unlike prior studies [80, 98, 61], no generator can easily match $\mathcal{D}_{ref}$ on our benchmark datasets. This confirms that the selected datasets offer a more informative and realistic assessment of generator capabilities.

## 4.2 Validity of Global Utility (Q2)

**Global utility serves as an effective metric for global structural fidelity.** Table 2 and Figure 2 (left) demonstrate a strong monotonic correlation between global utility and global CI scores

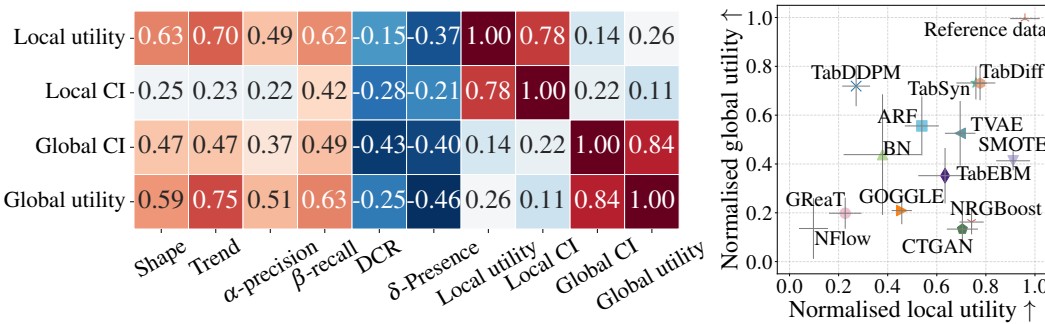

Figure 2: **Left:** Spearman's rank correlation heatmap based on metric values on six SCM datasets. Global utility correlates strongly with global CI, suggesting that global utility can effectively assess global structural fidelity as an SCM-free proxy of CI. **Right:** Mean normalised local utility vs. mean normalised global utility on 23 real-world datasets. SMOTE prioritises local utility, whereas TabDiff and TabSyn generally achieve a balanced preservation of both global and local data structures.

($r_s = 0.84, \mathrm{p} < 0.001$). In addition, global CI exhibits weaker correlations with all other metrics, showing conventional dimensions cannot sufficiently indicate structural fidelity. Appendix G further shows that global utility more closely approximates the ranking induced by global CI than local utility. In other words, generators that excel in capturing global structure also score highly in global utility (e.g., TabSyn and TabDiff), while the ones that struggle to capture global structure (e.g., NFlow and NRGBoost) perform poorly under global utility. Although global utility does not theoretically guarantee causal alignment, we note that even state-of-the-art causal discovery methods *cannot* theoretically ensure the inferred graphs align with the ground-truth SCMs [46]. In contrast, the empirical results validate the use of global utility for measuring global structural fidelity when the ground-truth SCM is unavailable.

**Local utility is not always the golden standard, due to its bias towards the local structure.** We further examine the correlation between local utility and local CI, which only considers the local structure associated with the prediction target. As shown in Figure 2 (left), local utility exhibits a strong correlation with local CI ($r_s = 0.78, \mathrm{p} < 0.001$), but a much weaker correlation with global CI ($r_s = 0.14, \mathrm{p} < 0.001$). The results indicate that local utility may reward "myopic" generators while missing the holistic data structure. This underscores the necessity of global utility for a more comprehensive evaluation of structural fidelity in tabular generation.

### 4.3 Structural Fidelity of Generators (Q3)

**Diffusion models generally capture the global structure well.** As reported in Table 2 and Figure 2 (right), diffusion-based models consistently achieve the highest scores in global structural fidelity: the Top-3 methods are TabDDPM, TabSyn, and TabDiff across both SCM and real-world datasets. We attribute their strong performance to the inherent capacity of diffusion models for learning permutation-invariant conditional distributions of each feature. At the training stage, since Gaussian noise is added independently to each feature, the diffusion network is optimised at every denoising step to reconstruct each feature by conditioning on all others. Consequently, it learns the conditional distribution for every feature $p(\boldsymbol{x}_j \mid \mathcal{X} \setminus \{\boldsymbol{x}_j\})$ simultaneously. Moreover, unlike autoregressive models, which generally rely on a fixed generation order, diffusion models impose no ordering constraint. This results in efficient computation (Figure 3) and permutation-invariant conditional distributions, a property that aligns naturally with the structure of tabular data. These theoretical properties align with the conditional independence analysis in Section 3.2.1, thus confirming that diffusion models are capable of capturing global structure.

**Interpolation and energy-based methods tend to prioritise local structure over global structure.** Figure 2 (right) shows that the interpolation method (e.g., SMOTE) and energy-based models (e.g., TabEBM and NRGBoost) can effectively capture local structure, yet perform poorly when modelling global structure. These two families of methods share a common trait in their generation process: they generate new samples from class-specific reference data. For example, in classification tasks, SMOTE interpolates between samples of the same class, and TabEBM samples from a class-specific energy surface. As a result, the generated samples are inevitably biased towards local structure.

**Structure learning methods struggle with tabular data generation.** One surprising finding is that BN and GOGGLE do not demonstrate strong performance in terms of structural fidelity, despite their

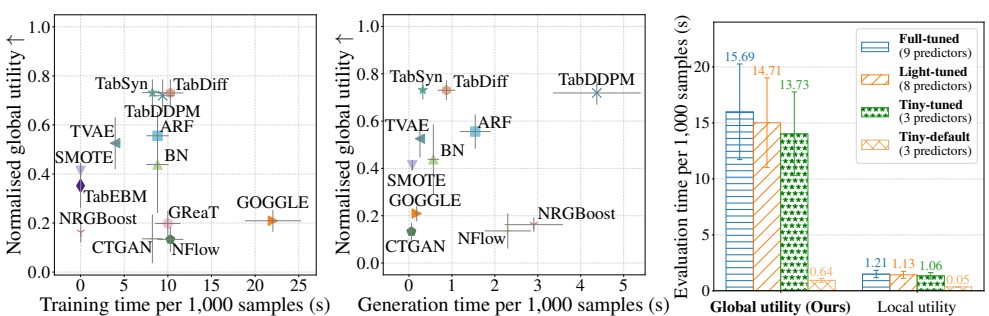

Figure 3: **Computation efficiency on 23 real-world datasets. Left:** Median training time per 1,000 samples vs. mean normalised global utility. **Middle:** Median generation time per 1,000 samples vs. mean normalised global utility. We exclude the outliers (TabEBM and GReaT) due to their long generation time (over 30s). Full results are in Appendix H. **Right:** Median evaluation time. Because global utility yields stable rankings across downstream predictors (Appendix G), computing global utility can be highly efficient with only a small ensemble of predictors (i.e., Tiny-default).

inductive bias towards learning tabular data structures. This observation aligns with prior work [89], which highlights that current causal discovery algorithms often struggle when the number of features exceeds 10. In contrast, our benchmark datasets have features from 7 up to 145. Furthermore, GOGGLE exhibits notable performance degradation when prior knowledge about the data structure is missing [58]. The results underscore the limitations of existing causal discovery methods in recovering precise causal structures from real-world data, further justifying our evaluation at the CPDAG level.

**Discussion on more models.** We further provide a detailed performance analysis of the remaining generators, such as autoregressive models, in Appendix G.

### 4.4 Practicability of Global Utility (Q4)

**Global utility is robust, stable, and efficient.** To evaluate the robustness of global utility, we examine how generator rankings vary under different configurations of downstream models. Appendix G shows that global utility yields stable rankings across both nine tuned predictors ("Full-tuned") and three untuned ones ("Tiny-default"). In contrast, local utility necessitates nine tuned predictors ("Full-tuned") for reliable results. In practice, we are often interested in identifying the most promising model before fine-tuning it for optimal performance [63]. As such, global utility offers an efficient and informative evaluation. As illustrated in Figure 3 (right), global utility ("Tiny-default") provides reliable rankings with a median runtime of just 0.64 seconds per 1,000 samples, nearly half the time required by local utility (1.21 seconds with "Full-tuned").

**Limitations and future work.** While our proposed global utility is a robust and effective metric for assessing global structural fidelity, it is an empirical approximation of the likely SCMs behind the data at hand. This is in line with several open challenges in the field, specifically the lack of causal discovery methods that can reliably infer the governing SCMs of real-world tabular data with strong theoretical guarantees [46, 89, 34, 71]. Addressing this gap would require advances in causal modelling, which we leave for future work. More discussion on future work is in Appendix G.

## 5 Conclusion

We present TabStruct, a principled benchmark for tabular data generators along with both structural fidelity and conventional dimensions. To address the challenge of assessing structural fidelity in the absence of ground-truth SCMs, we introduce global utility – a novel, SCM-free metric that enables unbiased and holistic evaluation for tabular data structure.

In our large-scale study of 13 generators across 29 datasets, we find that existing evaluation methods often favour models that capture local correlations while neglecting global structure. Our results show that diffusion models, due to their permutation-invariant generation process, offer valuable insights into the fundamental representation learning of tabular data. We further observe that the four evaluation dimensions are complementary, offering practical guidance for selecting suitable generators across diverse applications. TabStruct is an ongoing effort. As such, it will continue to evolve with additional datasets, generators, and evaluation metrics – both through our engagement and contributions from the community. We envision that the open-source nature of TabStruct will help drive progress in tabular generative modelling.

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
