# OpenReview forum: "TabStruct: Measuring Structural Fidelity of Tabular Data"
_NeurIPS.cc/2025/Datasets_and_Benchmarks_Track — Submitted to NeurIPS 2025 Datasets and Benchmarks Track_

### Official Review · Reviewer_cESG · 2025-06-25

**Rating:** 5
**Confidence:** 3

**Summary:**

This paper introduces a new evaluation framework that not only considers the common evaluation metrics but also the structural fidelity, which is rarely considered by previous works. To evaluate the structural fidelity, the paper uses the CI scores in concept. To overcome the difficulty that the calculation of CI scores require the knowledge of the ground-truth casual structure, which is hard to achieve for many real datasets, the paper proposes using the global utility as the practical implement.

The paper also conducts experiments in various generative models and datasets.

**Dataset Code Accessibility:**

Yes

**Dataset Code Comments:**

The dataset is public and the authors have clearly written the sources of them in appendix E.

**Ethical Considerations:**

No, there are no or only very minor ethics concerns

**Final Justification:**

My concerns are addressed, therefore I've decided to raise my score to 5.

**Limitations Weaknesses:**

If I understand correctly, the paper CauTabBench's idea is to use the generated synthetic data to test the model structure's ability of capturing the casual relationship. The reason why they do this is that they want to test the model's ability in data with ground-truth casual structure. Their method can in fact be generalized to higher dimensions easily, so I feel that it is not fair to criticize that their paper is limited to toy datasets.

To give a more comprehensive comparison, I suggest to further investigate whether the global utility is more correlated with the CI scores compared to the metric proposed in the CauTabBench paper. That is, you can consider evaluating the model's ability in some datasets \{A\}, using the metrics of both global utility and the one proposed in CauTabBench, then calculating the model's CI scores in some other datasets \{B\}, to see whether the result provided by the global utility or the CauTabBench paper's metric is more correlated with the CI scores calculated in \{B\}.

Also, is it possible to provide some theoretical guarantee or some theoretical intuition that the global utility is a good measurement of the structural fidelity?

**Strengths Contributions:**

The motivation of this paper is clear and important: The evaluation of the structural fidelity is a very important problem, while rarely considered by previous works. As discussed in the paper, CauTabBench is the only paper that considers the structural fidelity explicitly.

The idea of using the CI scores to evaluate the structural fidelity and the usage of the global utility in practical implement make a lot of sense.

The experiments are comprehensive and support the claims in the paper, especially the important one that the global utility is highly consistent with the CI scores.

---

> ### Author Rebuttal · Authors · 2025-07-30
>
> Thank you for the positive feedback! We address all of your concerns and comments below (Table R13). We will update the manuscript accordingly to include the additional clarifications and full results of new experiments.
>
> *Table R13:* Number of our responses to the reviewers’ comments (`#Raised`/`#Replied`).
>
> | Reviewer | # Limitations/Weaknesses | # New experiments |
> | --- | --- | --- |
> | `cESG` | 3/3 | 1 |
>
> > W1: it is not fair to criticize that their paper is limited to toy datasets
> >
>
> Thank you for the detailed clarifications. We fully acknowledge the valuable insights and contributions of CauTabBench.
>
> **First**, we would like to clarify that the term “toy dataset” (Line 41) is intended to broadly refer to datasets that are *not* derived from real-world sources. Both the synthetic datasets used in CauTabBench and the SCM datasets in TabStruct fall into this category. In other words, the term does *not* necessarily imply that the datasets are low-dimensional.
>
> **Second**, while we agree that the CauTabBench framework is, in principle, scalable to high-dimensional datasets, we note that the specific causal discovery methods it employs may not be practically scalable. For instance, prior work `[1, 8, 9]` has demonstrated the high computational costs associated with causal discovery algorithms such as PC. Empirically, we find that the vanilla PC algorithm in CauTabBench can require up to 168 hours (i.e., 7 days) to complete on datasets with more than 50 features. As a result, many of the metrics used in CauTabBench may be computationally impractical for the real-world datasets considered in TabStruct, indicating that CauTabBench would require additional technical optimisations for practical deployment. Moreover, recent work `[3]` shows that even state-of-the-art methods often perform poorly on real-world data and may mislead users. Therefore, applying CauTabBench to real-world datasets would pose challenges in terms of both scalability and reliability.
>
> **Third**, as you, we also strongly believe that precision in presentation is a crucial aspect of scientific communication, and as such, we will modify the writing accordingly to further highlight the contributions of CauTabBench on datasets with known causal structures and the distinct novelties introduced by TabStruct.
>
> > W2: whether the global utility is more correlated with the CI scores compared to the metric proposed in the CauTabBench paper
> >
>
> Below, we report the correlation between global CI and TabStruct/CauTabBench’s metrics. We will add the full results to the revised manuscript.
>
> **Experimental design**
>
> - Step-1: We construct two dataset collections, A and B, each containing three SCM datasets. Specifically, `A={Hailfinder, Insurance, MEHRA}` and `B={Sangiovese, Healthcare, MAGIC-IRRI}`.
> - Step-2: On datasets in A, we compute our global utility as well as three CauTabBench metrics: skeleton-F1, direction-ACC, and direction-F1.
> - Step-3: On datasets in B, we compute the global CI.
> - Step-4: We compute Spearman’s rank correlation between global CI and the other metrics.
>
> **Experimental setup**
>
> - All settings (13 generators, data splitting, etc.) are the same as Section 4.
>
> **Results**
>
> - **Important:** Table R14 shows that global utility exhibits a substantially stronger correlation with global CI compared to the CauTabBench metrics. This suggests that global utility can provide a robust, SCM-free approach for assessing global structural fidelity.
> - The relatively weaker correlation of CauTabBench metrics is likely due to their dependence on causal discovery algorithms. For instance, skeleton-F1 uses PC algorithm to recover causal graphs from synthetic tabular data. However, PC algorithms could suffer notable performance degradation as the number of features increases `[8, 9]`. This aligns with the broader limitations of existing causal discovery methods on tabular datasets `[2, 3]`.
>
> *Table R14:* Spearman’s rank correlation between global CI (computed on B) and other metrics (computed on A) (p<0.001).
>
> |  | Global CI $\uparrow$ |
> | --- | --- |
> | **CauTabBench metrics** |  |
> | skeleton-F1 | 0.42 |
> | direction-ACC | 0.44 |
> | direction-F1 | 0.44 |
> | **TabStruct metric** |  |
> | **Global utility (Ours)** | **0.81** |
>
> > W3: theoretical guarantee or some theoretical intuition that the global utility is a good measurement of the structural fidelity
> >
>
> **First**, we would like to emphasise that developing a theoretically provable structural fidelity metric for *real-world tabular data* is highly challenging, even infeasible. In most practical cases, ground-truth causal structures are unavailable, precluding the possibility of theoretical validation. To establish such validation would require a reliable causal discovery algorithm capable of recovering high-quality SCM from observed data. Despite substantial research efforts, recent studies `[2, 3]` demonstrate that even state-of-the-art causal discovery methods – while theoretically sound under certain assumptions – still perform poorly when applied to real-world data, even leading to misleading guidance for users. The underperformance further indicates a gap between theoretical causal ML frameworks and the complexities of real-world tabular data. Following prior studies on tabular foundation models `[4, 5]`, TabStruct adopts an *“SCM-inspired and real-data-validated”* paradigm to mitigate the misalignment between theory and practice, providing users with actionable and empirically grounded insights into tabular data generation.
>
> **Second**, that said, we provide a three-step chain of theoretical motivations and quantitative validations to support the alignment between global utility and global structural fidelity. For clarity, we summarise the key components below:
>
> - **Step-1 Decomposition** (Lines 150-160): We split structural fidelity metrics into local CI (an individual variable) and global CI (across all variables).
> - **Step-2 Local CI vs. Local utility** (Lines 164-170): Achieving high local CI requires the generator to correctly identify (un-)informative features under various d-separation/connection sets. This aligns closely with Markov blanket based feature selection `[6]`. Importantly, correctly identifying (un-)informative features generally improves downstream predictive performance `[7]`. The strong correlation between local CI and local utility further validates their connections.
> - **Step-3 Aggregation to global level** (Lines 195-199): Building on the heuristics from Step-2, we examine the relationship between global CI and global utility.  Our evaluation spans 13 generators across six SCM datasets, one of the most comprehensive to date (Table 1 and Appendix B.4). The results consistently show a strong correlation between global CI and global utility, in line with the insights derived in the above steps.
>
> **Third**, we will make the TabStruct library freely available, and its development will be an ongoing, community-driven endeavour. Therefore, TabStruct will continue to evolve with advances in causal modelling. We believe that the open-source nature of TabStruct will help drive progress in bridging the gap between theoretical foundations and real-world tabular data challenges.
>
> **References**
>
> - `[1]` Kaddour, Jean, et al. "Causal machine learning: A survey and open problems." arXiv:2206.15475 (2022).
> - `[2]` Zanga, Alessio, et al. "A survey on causal discovery: theory and practice." IJAR 2022.
> - `[3]` Nastl, Vivian, and Moritz Hardt. "Do causal predictors generalize better to new domains?." NeurIPS 2024.
> - `[4]` Qu, Jingang, et al. "TabICL: A Tabular Foundation Model for In-Context Learning on Large Data." ICML 2025.
> - `[5]` Hollmann, Noah, et al. "Accurate predictions on small data with a tabular foundation model." Nature 2025.
> - `[6]` Fu, Shunkai, and Michel C. Desmarais. "Markov blanket based feature selection: a review of past decade." Proceedings of the world congress on engineering 2010.
> - `[7]` Cherepanova, Valeriia, et al. "A performance-driven benchmark for feature selection in tabular deep learning." NeurIPS 2023.
> - `[8]` Le, Thuc Duy, et al. "A fast PC algorithm for high dimensional causal discovery with multi-core PCs." TCBB 2016.
> - `[9]` Shiragur, Kirankumar, et al. "Causal discovery with fewer conditional independence tests." ICML 2024.

---

> > ### Comment · Reviewer_cESG · 2025-08-05
> >
> > Thank you for the rebuttal, which addresses my concerns. I'll raise my score to 5.

---

### Official Review · Reviewer_wm4v · 2025-06-28

**Rating:** 4
**Confidence:** 4

**Summary:**

In this paper, the authors propose TabStruct, a benchmark for evaluating synthetic tabular data generators with a focus on structural fidelity—that is, preserving the dependencies among features that existing benchmarks have not adequately measured. To make this well-defined, they introduce Conditional Independence (CI)-based metrics for datasets with known Structural Causal Models (SCMs), and propose global utility as an SCM-free metric to assess structural fidelity when no causal graph is available. Based on extensive comparisons across different methods, they find that diffusion models, due to their permutation-invariant generation process, offer valuable advantages for capturing the fundamental dependency structure in tabular data.

**Dataset Code Accessibility:**

Yes

**Ethical Considerations:**

No, there are no or only very minor ethics concerns

**Final Justification:**

The overall experiments and writing meet the expectations of a conference paper, and the problem is well-motivated. The authors have put in a solid effort to improve the paper in response to the reviews. Therefore, I believe the current score is reasonable and will remain unchanged.

**Limitations Weaknesses:**

**[Reliance on the Observed Conditional Dependences]** In this paper, the structural fidelity metrics for both CI-based and global utility-based metrics all assume that the observed conditional independences in the data reflect the true underlying structure. However, in the real world, datasets often contain unobserved confounders. This would induce spurious dependencies that are faithfully reproduced by the generators, but it is not the actual causal mechanism between the features. Therefore, I am wondering if some of the experiments/simulations need to be done to check how the hidden confounding could bias the evaluations and how the users should interpret the fidelity scores in that setting.

**[Unbalanced influence of the Global Utility Metric]** In the paper, the global utility metric relies on predictive performance, which introduces potential bias toward variables with strong predictive signals. In practice, features with high variable importance can dominate the global utility score, while weaker but still meaningful dependencies may contribute little. As a result, the evaluation may disproportionately reward generators that capture only the strongest effects while failing to assess fidelity for subtler relationships in the data. I am wondering if the authors could discuss this problem further, as it has important implications for interpreting global utility scores, particularly in settings with diverse or imbalanced feature importance.

**[Adding a Causal Inference Check]** Given that the central idea of this work is to evaluate how well synthetic data preserves the underlying structural causal model (SCM), I am wondering if the authors would like to include some validation or discussion connecting these structural fidelity metrics to causal inference tasks. For example, assessing whether synthetic data supports accurate estimation of treatment effects or counterfactual predictions could help demonstrate the practical relevance of preserving these dependencies and improve readers’ understanding of the metric’s implications. Additionally, from the perspective of the global utility metric, proposed as an SCM-free proxy for structural fidelity, it would be interesting for the paper to discuss whether high global utility scores can intuitively indicate support for valid marginal causal effect estimation or reasoning. Therefore, I think building a connection between causal inference and the global utility score would definitely strengthen the paper.

**Strengths Contributions:**

**[Identification of structure fidelity problem]** The authors identify an important gap in current evaluation practice: existing benchmarks focus on density estimation, ML task accuracy, or privacy preservation, but rarely test whether synthetic data preserves inter-feature dependencies that can be crucial for scientific validity or downstream analysis.

**[Comprehensive Experiments Support]** The paper conducts extensive experiments across a wide range of models and datasets, including both synthetic and real-world settings. It provides comprehensive, well-designed figures and analyses that convincingly illustrate the importance of evaluating structural fidelity in synthetic data generation.

---

> ### Author Rebuttal · Authors · 2025-07-30
>
> Thank you for the positive feedback! We address all of your concerns and comments below (Table R7). We will update the manuscript accordingly to include the additional clarifications and full results of new experiments.
>
> *Table R7:* Number of our responses to the reviewers’ comments (`#Raised`/`#Replied`).
>
> | Reviewer | # Limitations/Weaknesses | # New experiments |
> | --- | --- | --- |
> | `wm4v` | 3/3 | 3 |
>
> > W1: Reliance on the Observed Conditional Dependences
> >
>
> Thank you for your insightful comment. Below, we present a proof-of-concept experiment designed to investigate the behaviour of global utility on datasets with unobserved confounders. We will add the full results to the revised manuscript.
>
> **Experimental design**
>
> - Step-1: Given an SCM, we inject $n_\text{conf}$ exogenous latent confounders $U=\\{U_1, \ldots, U_{n_{\text {conf}}}\\}$. Each $U_k \sim N(0,1)$ is associated with a child set $S_k \subset \mathcal{X}$ with $\left|S_k\right|=m_k \geq 2$. For each variable $x_j \in S_k$ , we update its structure function via $x_j =f_j(\text{pa}(x_j), \epsilon_j)+\sum_{k:j \in S_k} \lambda_{kj}U_k$, where $U_k$ are exogenous and mutually independent, and $\lambda_{kj} \sim \text{Unif}(0.5, 1.5)$ is the confounding strength. This step yields a new causal graph $\mathcal{G}_{\text{DAG}}^{\text {Conf}}$.
> - Step-2: We marginalise out the unobserved confounders to transform $\mathcal{G}^\text{Conf}\_\text{DAG}$ into a maximal ancestral graph (MAG) $\mathcal{G}_\text{MAG}$.
> - Step-3: On $\mathcal{G}_\text{MAG}$, we derive the conditional independence relationships with m-separation.
> - Step-4: Following Equation 3, we compute “Global CI (MAG)”, a variant of the normal global CI, with the derived CI relations in Step-3.
> - Repeat the above four steps for three SCM datasets: Sangiovese, Healthcare, and MEHRA.
>
> **Experimental setup**
>
> - We perform a grid search over the hyperparameters in Step-1, resulting in 15 runs per dataset for each data split:
>     - $m_k \in\\{2,3,4\\}$
>     - $n_\text{conf} \in \\{1, 2, 3, 4, 5\\}$
> - We use the same conditional independence tests as Appendix C.1.2.
> - All settings (data splitting, etc.) are the same as Section 4.
>
> **Results**
>
> - **Important:** Global utility maintains stable performance (Table R8) and high correlation with global CI (MAG) (Table R9) across varying numbers of unobserved confounders. This highlights the robustness and generalisability of global utility.
> - These results are promising for extending global utility to applications involving datasets with unobserved confounders.
>
> *Table R8:* Benchmark results on three SCM datasets with unobserved confounders, aggregated over 13 tabular generators.
>
> | # unobserved confounders | Global CI (MAG) | Global utility (Ours) | Local CI (MAG) | Local utility |
> | --- | --- | --- | --- | --- |
> | 1 | 0.80 | 0.79 | 0.84 | 0.87 |
> | 2 | 0.80 | 0.77 | 0.84 | 0.84 |
> | 3 | 0.75 | 0.79 | 0.81 | 0.85 |
> | 4 | 0.74 | 0.75 | 0.80 | 0.84 |
> | 5 | 0.71 | 0.77 | 0.79 | 0.82 |
>
> *Table R9:* Spearman’s rank correlation with global CI (MAG) on three SCM datasets with unobserved confounders (p<0.001).
>
> | # unobserved confounders | Global utility (Ours) | Local CI (MAG) | Local utility | Shape | Trend | $\alpha$-precision | $\beta$-recall | DCR | $\delta$-presence |
> | --- | --- | --- | --- | --- | --- | --- | --- | --- | --- |
> | 1 | **0.76** | 0.22 | 0.14 | 0.26 | 0.30 | 0.41 | 0.24 | -0.28 | -0.23 |
> | 2 | **0.72** | 0.14 | 0.18 | 0.22 | 0.30 | 0.41 | 0.26 | -0.17 | -0.16 |
> | 3 | **0.76** | 0.18 | 0.19 | 0.22 | 0.31 | 0.40 | 0.27 | -0.25 | -0.16 |
> | 4 | **0.75** | 0.15 | 0.19 | 0.22 | 0.31 | 0.39 | 0.25 | -0.26 | -0.15 |
> | 5 | **0.72** | 0.16 | 0.16 | 0.25 | 0.33 | 0.35 | 0.26 | -0.28 | -0.16 |
>
> > W2: Unbalanced influence of the Global Utility Metric
> >
>
> **(1) Empirical justification.** Global utility offers a well-balanced evaluation, as it correlates strongly with global CI. By design, global CI treats all conditional independence relations uniformly and does not bias towards any specific feature.  If global utility were disproportionately dominated by a small subset of features, its correlation with global CI would be weakened. However, the observed strong empirical correlation indicates that global utility is not heavily skewed by any particular feature.
>
> **(2) Conceptual clarification**. For clarity, we note that global utility is *not* computed by directly aggregating raw predictive scores across features. Instead, the utility per variable is first normalised by the corresponding reference performance before aggregation (Equation 4). This normalisation ensures global utility captures *relative* rather than *absolute* predictive powers, thereby mitigating column-specific bias. This design choice is consistent with established practices in multi-dataset evaluation, where ADTM is widely used to avoid dataset-specific bias `[1, 2]`.
>
> **(3) Impact of column bias mitigation.** To assess the effect of normalisation, we compare global utility computed using *absolute* predictive scores versus *relative* (i.e., normalised) scores. As shown in Table R10, using unnormalised scores leads to a substantially weaker correlation with global CI. Specifically, Spearman’s $\rho$ drops from 0.84 to 0.57. This finding supports that the normalisation design plays an important role in mitigating feature dominance and enabling structural fidelity assessment.
>
> *Table R10:* Spearman’s rank correlation between global CI and global utility across six SCM datasets (p<0.001).
>
> |  | Global CI $\uparrow$ |
> | --- | --- |
> | Global utility (absolute performance) | 0.57 |
> | **Global utility (relative performance)** | **0.84** |
>
> > W3: Adding a Causal Inference Check
> >
>
> We add the causal inference check for 13 tabular data generators across six SCM datasets. Following Chen et al. `[3]` and CauTabBench `[4]`, we assess performance by learning SCMs from synthetic data and comparing them against the ground-truth SCMs on both interventional and counterfactual inference tasks. We will add the full results to the revised manuscript.
>
> **Experimental design**
>
> - Step-1: Given a ground-truth SCM $M_\text{ref} = \langle \mathcal{X}, \mathcal{G}\_\text{ref}, \mathcal{F}\_\text{ref}, \mathcal{E}\_\text{ref} \rangle$ (see Lines 127-132 for a detailed definition), we obtain an SCM $M_\text{syn} = \langle \mathcal{X}, \mathcal{G}\_\text{ref}, \mathcal{F}\_\text{syn}, \mathcal{E}\_\text{syn} \rangle$ via applying DoWhy `[5]` on synthetic data. This involves learning new structural functions $\mathcal{F}\_\text{syn}$ and exogenous noise $\mathcal{E}\_\text{syn}$ from the synthetic data, while keeping the causal graph $\mathcal{G}\_\text{ref}$ fixed.
> - Step-2: For interventional evaluation, we perform 10 interventions per variable, then generating 100,000 interventional samples per intervention under $M_\text{ref}$ and $M_\text{syn}$. Then, we compute intervention mean squared error (I-MSE) by comparing the expected values of the remaining variables.
> - Step-3: For counterfactual evaluation, given observed data, we apply 10 interventions per variable and generate 100,000 counterfactual samples using both $M_\text{ref}$ and $M_\text{syn}$. We compute the mean counterfactual values from both SCMs and compute the counterfactual mean squared error (C-MSE).
>
> **Experimental setup**
>
> - All settings (data splitting, ADTM normalisation, etc.) are the same as Section 4. The reported results are averaged over 10 runs.
>
> **Experimental result**
>
> - **Important:** As shown in Table R11, the Top-5 performant generators in causal inference check (TabSyn, TabDDPM, TabDiff, TVAE, and ARF) are consistent with those in both global CI and global utility. This further supports the global utility’s ability to indicate the global causal structure of tabular data.
> - Table R12 further shows that existing evaluation metrics show considerably weaker correlations with causal inference performance, while global utility serves as an effective indicator.
>
> *Table R11:* Causal inference performance of 13 tabular generators on six SCM datasets.
>
> | Generator | I-MSE $\downarrow$ | C-MSE $\downarrow$ |
> | --- | --- | --- |
> | SMOTE | 0.32 | 0.45 |
> | BN | 0.37 | 0.51 |
> | TVAE | 0.16 | 0.27 |
> | GOGGLE | 0.59 | 0.75 |
> | CTGAN | 0.87 | 0.90 |
> | NFlow | 0.97 | 0.98 |
> | ARF | 0.12 | 0.23 |
> | TabDDPM | 0.10 | 0.22 |
> | TabSyn | **0.09** | **0.20** |
> | TabDiff | **0.09** | 0.21 |
> | TabEBM | 0.34 | 0.47 |
> | NRGBoost | 0.55 | 0.70 |
> | GReaT | 0.59 | 0.75 |
>
> *Table R12:* Spearman’s rank correlation on six SCM datasets (p<0.001).
>
> |  | Global utility (Ours) | Shape | Trend | $\alpha$-precision | $\beta$-recall | DCR | $\delta$-presence | Local utility | Local CI | Global CI |
> | --- | --- | --- | --- | --- | --- | --- | --- | --- | --- | --- |
> | I-MSE $\downarrow$ | **-0.90** | -0.32 | -0.33 | -0.35 | -0.19 | 0.23 | 0.15 | -0.21 | -0.17 | -0.80 |
> | C-MSE $\downarrow$ | **-0.83** | -0.17 | -0.40 | -0.16 | -0.14 | 0.36 | 0.21 | -0.45 | -0.24 | -0.82 |
>
> **References**
>
> - `[1]` Grinsztajn, Léo, et al. "Why do tree-based models still outperform deep learning on typical tabular data?." NeurIPS 2022.
> - `[2]` McElfresh, Duncan, et al. "When do neural nets outperform boosted trees on tabular data?." NeurIPS 2023.
> - `[3]` Chen, Asic, et al. "Structured neural networks for density estimation and causal inference." NeurIPS 2023.
> - `[4]` Tu, Ruibo, et al. "Causality for tabular data synthesis: A high-order structure causal benchmark framework." arXiv:2406.08311 (2024).
> - `[5]` Sharma, Amit, and Emre Kiciman. "DoWhy: An end-to-end library for causal inference." arXiv:2011.04216 (2020).

---

> > ### Comment · Reviewer_wm4v · 2025-08-05
> >
> > Thank the authors for all the work that have been done for the rebuttal. The extra experiments and explanations added would definitely strengthen the paper. I have no further concern.

---

### Official Review · Reviewer_WDBm · 2025-06-30

**Rating:** 5
**Confidence:** 3

**Summary:**

The paper introduces a new benchmark and evaluation framework for tabular data generators. The authors argue that existing evaluation methods are inadequate because they largely ignore "structural fidelity", which is described as the extent to which synthetic data captures the underlying causal relationships (Structural Causal Models, or SCMs) of the real data. They argue that popular metrics like ML efficacy are biased towards specific prediction targets and fail to provide a holistic assessment of a generator's quality.
The main contributions of this work are:
1. A novel evaluation framework
2. The "Global Utility" metric
3. The TabStruct benchmark  and an open-source library
4. Empirical findings

**Dataset Code Accessibility:**

Yes

**Dataset Code Comments:**

good quality, clear material overall

**Ethical Considerations:**

No, there are no or only very minor ethics concerns

**Final Justification:**

Good, solid work. The details in the rebuttal should be reflected in the paper

**Limitations Weaknesses:**

W1.
The central claim of the paper is that global utility serves as an effective proxy for structural fidelity. While the authors provide empirical evidence for this, this remains an empirically observed correlation rather than a theoretical necessity. Authors acknowledge this limit, stating that "global utility does not theoretically guarantee causal alignment"
A generator could be engineered to score highly on global utility (i.e., by being good at one-vs-rest prediction for all columns) without really  capturing the true conditional independence structure defined by SCM.
Also, the use of AutoGluon's performance in the intermediary step leads to a possible interpretation of the global utility as a measure of how well an ensemble of ML models predict each variable, which is not necessarily the same as preserving the true data-generating process.
Authors could strengthen their claims by discussing the theoretical conditions under which global utility would be expected to align with true structural fidelity.

W2.
Despite including autoregressive models in the evaluation, their performance analysis is brief and somewhat dismissive.
Autoregressive models explicitly model conditional distributions, which is related to structural fidelity. Given their natural fit for modeling dependencies, why they underperform in capturing global structure? Is it the fixed-ordering bias? Error accumulation? The paper would benefit from a comparative analysis against this strong baseline family.

W3.
The proposed global utility metric, while powerful, appears computationally intensive. The evaluation involves training an ensemble of nine predictors for every single column in the dataset.
The "Full-tuned" evaluation, required for robust local utility results, takes over 15 seconds. For datasets with many columns (e.g., hundreds) or large sample sizes, the total evaluation time could become prohibitive.
Authors should be more explicit about the scalability of their evaluation pipeline. They could provide a complexity analysis of computing global utility with respect to the number of samples (N), features (D), and the number of ensemble predictors (K).
Can they also explore and recommend more lightweight but still reliable configurations for the downstream prediction? Will the stability of the rankings degrade?

W4.
The six expert-validated SCM datasets may not fully represent the complexity or types of structures found in large-scale tabular data problems.They  have a limited number of features (7-64) and the correlation observed between global utility and global CI may not hold  for datasets with hundreds or thousands of features, where causal discovery itself becomes very difficult.
Is this a potential limitation in dataset scope to discuss? Otherwise, authors should include results on larger or more complex synthetic SCMs. As expert-validated SCMs are rare, they could generate larger SCMs using established procedures (e.g., random DAG generation) and treat the generating graph as the ground truth.

**Strengths Contributions:**

S1
principled and novel evaluation framework that positions the assessment of tabular data generators around the concept of structural fidelity. The authors argue that we should not reward models that capture narrow, task-specific correlations while failing to preserve the holistic, underlying causal structure of the data.
To make this concept practical, they introduce global utility. The proposed metric correlates strongly with ground-truth structural fidelity on expert-validated SCM datasets, bringing credibility to its use on real-world data.

S2
The new framework is instantiated in the TabStruct benchmark, covering 13 modern generators across 29 curated datasets. The authors demonstrate  experimental rigor and draw actionable insights, such as identifying the general superiority of diffusion models in capturing global data structure.

---

> ### Author Rebuttal · Authors · 2025-07-30
>
> Thank you for the positive feedback! We address all of your concerns and comments below (Table R4). We will update the manuscript accordingly to include the additional clarifications and full results of new experiments.
>
> *Table R4:* Number of our responses to the reviewers’ comments (`#Raised`/`#Replied`).
>
> | Reviewer | # Limitations/Weaknesses | # New experiments |
> | --- | --- | --- |
> | `WDBm` | 4/4 | 2 |
>
> > W1: Global utility … an empirically observed correlation rather than a theoretical necessity.
> >
>
> **First**, developing a theoretically provable structural fidelity metric for *real-world tabular data* is highly challenging, as ground-truth causal structures are rarely available, even precluding the possibility of theoretical validation. Therefore, establishing such validation would require a reliable causal discovery algorithm. Despite substantial research efforts, recent work `[1, 2]` shows that even state-of-the-art causal discovery methods often perform poorly on real-world data and may mislead users. This highlights a gap between theoretical causal ML frameworks and the complexities of real-world tabular data. Following prior studies on tabular foundation models `[3, 4]`, TabStruct adopts an *“SCM-inspired and real-data-validated”* paradigm to mitigate such misalignment, providing users with actionable and empirically grounded insights into tabular data generation.
>
> **Second**, that said, we provide a three-step chain of theoretical motivations and quantitative validations to support the alignment between global utility and global structural fidelity. For clarity, we summarise the key components below:
>
> - **Step-1 Decomposition** (Lines 150-160): We split structural fidelity metrics into local CI (an individual variable) and global CI (across all variables).
> - **Step-2 Local CI vs. Local utility** (Lines 164-170): Achieving high local CI requires the generator to correctly identify (un-)informative features under various d-separation/connection sets. This aligns closely with Markov blanket based feature selection `[5]`. Importantly, correctly identifying (un-)informative features generally improves downstream predictive performance `[6]`. The strong correlation between local CI and local utility further validates their connections.
> - **Step-3 Aggregation to global level** (Lines 195-199): Building on the heuristics from Step-2, we examine the relationship between global CI and global utility.  Our evaluation spans 13 generators across six SCM datasets, one of the most comprehensive to date (Table 1 and Appendix B.4). The results consistently show a strong correlation between global CI and global utility, in line with the insights derived in the above steps.
>
> **Third**, TabStruct is designed to evaluate structural fidelity at the CPDAG level, where an observed dataset may correspond to multiple Markov-equivalent SCMs. In Appendix D.2, we discuss this design choice. Here, we highlight the key points for clarity:
>
> - **Important:** CPDAG-level evaluation balances efficiency with the validity of the underlying causal structures.
> - Skeleton has a limited capacity due to the loss of all causal directions.
> - DAG level may provide misleading guidance for users due to the lack of reliable causal discovery methods for real-world datasets.
>
> Therefore, while we agree that the “one-vs-rest generator” may exist, this still implies that the generator captures a Markov-equivalent SCM to the ground truth – a meaningful achievement in the tabular data generation. For instance, Markov equivalence ensures consistency with the ground-truth physical laws (Figure 1). In contrast, existing evaluation dimensions do not explicitly account for CPDAG-level structural fidelity, often favouring interpolation-based methods that may overlook causal structures.
>
> > W2: Why do autoregressive models underperform in capturing global structure?
> >
>
> In Appendix G.2, we discuss the limitations of autoregressive tabular generators. Here, we summarise the key points for clarity:
>
> - **Important:** We agree that the misaligned bias in column ordering is a primary factor for the suboptimal performance of autoregressive tabular generators.
> - Autoregressive generators model the data distribution by linearising features according to a column order $\pi$. For tabular data, the ideal order $\pi^\*$ aligns with the topological order from the true SCM. However, since $\pi^*\$ is usually unavailable in practice, using a mismatched $\pi$ can harm structural fidelity. While prior work `[7, 8]` improves robustness by finetuning LLMs on random orders, this approach does not explicitly align models with the dataset's true causal structure.
>
> To further explore the effect of directional bias, we conduct a proof-of-concept experiment using six SCM datasets (Table R5). Specifically, we introduce *GReaT-sort*, a variant of GReaT finetuned solely using the ground-truth topological order derived from each SCM. We will add the full results to the revised manuscript.
>
> **Experimental setup**
>
> - All settings (data splitting, etc.) are the same as Section 4. The reported results are averaged over 10 runs.
> - GReaT-sort and GReaT share identical model configurations, and the only difference lies in the column order used during finetuning.
>
> **Results**
>
> - **Important:** GReaT-sort consistently outperforms GReaT by a clear margin across all datasets, showing that the mismatched bias in column ordering constrains the performance of autoregressive tabular generators.
>
> *Table R5:* Global utility of GReaT and GReaT-sort on six SCM datasets.
>
> |Generator| Hailfinder | Insurance | Sangiovese | Healthcare | MAGIC-IRRI | MEHRA |
> | --- | --- | --- | --- | --- | --- | --- |
> | GReaT | 0.29 | 0.32 | 0.31 | 0.19 | 0.41 | 0.26 |
> | **GReaT-sorted** | **0.43** | **0.49** | **0.40** | **0.43** | **0.49** | **0.31** |
>
> > W3: Can they also explore and recommend more lightweight but still reliable configurations for the downstream prediction? Will the stability of the rankings degrade?
> >
>
> In short – yes, researchers and practitioners can safely use the “Tiny-default” configuration to compute global utility, taking only **0.64s** per 1000 samples, while maintaining reliable and consistent generator rankings.
>
> **First**, we clarify the differences between local and global utility, which leads to distinct requirements for predictor configurations (Section 4.4 and Appendix G.3):
>
> - Local utility focuses solely on the predictive performance of the target variable $y$. Its column-specific bias leads to unstable generator rankings across predictor configurations (Figure 7).
> - Global utility accounts for all variables, thereby mitigating the column-specific bias. Across large-scale evaluations on 23 real-world datasets, “Tiny-default” yields the same generator rankings as “Full-tuned”.
>
> **Second**, due to the instability of the local utility, users have to use the “Full-tuned” configuration for reliable results. In contrast, the robustness of global utility allows safe use of the efficient “Tiny-default”. In other words, global utility reduces tuning burden and dependency on predictor count while still producing stable rankings. Concretely, computing global utility with “Tiny-default” takes only *0.64s (not 15.69s)* per 1000 samples, while computing local utility requires nearly double the time (1.21s) for comparable reliability.
>
> **Third**, global utility, implemented with the lightweight “Tiny-default” configuration, is a practically valuable and computationally manageable metric for researchers and practitioners.
>
> > W4: include results on larger or more complex synthetic SCMs
> >
>
> Thank you for acknowledging the scarcity of expert-validated SCMs. We have made a concerted effort to include SCM datasets that closely match the structures of real-world data.
>
> Below, we report the correlation between global CI and other metrics on five more complex SCMs (Table R6). We will add the full results to the revised manuscript.
>
> **Experimental setup**
>
> - We select five large SCM datasets (ANDES, Diabetes, Link, Pathfinder, and PIGS) from the bnlearn repository `[9]`, which contain 100,000 samples with 109-724 features.
> - All settings (data splitting, etc.) are the same as Section 4.
>
> **Results**
>
> - Global utility exhibits a strong correlation across both the large SCM datasets and the six smaller ones in the main text. This provides additional empirical evidence that global utility reliably reflects global structural fidelity of tabular data.
>
> *Table R6:* Spearman’s rank correlation with global CI on SCM datasets (p<0.001).
>
> | |Global utility (Ours)|Shape|Trend|$\alpha$-precision|$\beta$-recall|DCR|$\delta$-presence|Local utility|Local CI|
> |---| --- | --- | --- | --- | --- | --- | --- | --- | --- |
> | 5 large SCM datasets | **0.81** | 0.26 | 0.24 | 0.31 | 0.33 | -0.25 | -0.22 | 0.13 | 0.11 |
> | 11 SCM datasets (6 in main text + 5 large) | **0.83** | 0.40 | 0.30 | 0.36 | 0.46 | -0.30 | -0.26 | 0.13 | 0.19 |
>
> **References**
>
> - `[1]` Nastl, Vivian, et al. "Do causal predictors generalize better to new domains?." NeurIPS 2024.
> - `[2]` Zanga, Alessio, et al. "A survey on causal discovery: theory and practice." IJAR 2022.
> - `[3]` Hollmann, Noah, et al. "Accurate predictions on small data with a tabular foundation model." Nature 2025.
> - `[4]` Qu, Jingang, et al. "TabICL: A Tabular Foundation Model for In-Context Learning on Large Data." ICML 2025.
> - `[5]` Fu, Shunkai, et al. "Markov blanket based feature selection: a review of past decade." PWCE 2010.
> - `[6]` Cherepanova, Valeriia, et al. "A performance-driven benchmark for feature selection in tabular deep learning." NeurIPS 2023.
> - `[7]` Borisov, Vadim, et al. "Language Models are Realistic Tabular Data Generators." ICLR 2023.
> - `[8]` Fang, Xi, et al. "Large Language Models (LLMs) on Tabular Data: Prediction, Generation, and Understanding-A Survey." TMLR 2024.
> - `[9]` Marco Scutari. bnlearn-an r package for bayesian network learning and inference. UCL Genetics 2011.

---

> > ### Comment · Reviewer_WDBm · 2025-08-05
> >
> > Thanks for your rebuttal. I will increase my score

---

### Official Review · Reviewer_8PZA · 2025-07-02

**Rating:** 3
**Confidence:** 4

**Summary:**

The paper presents TabStruct, a new benchmark framework for tabular data evaluation. Specifically, TabStruct introduces two new evaluation metrics that focus on evaluating structural fidelity: conditional independence (CI) and global utility, where CI utilizes statistical tests to measure how well the synthetic data maintains the independence of features when the ground truth structural causal model (SCM) is present, and uses global utility as a proxy for global CI on real-world datasets where the SCM is absent. Experimental results show that there is a high correlation between global CI and global utility, indicating the effectiveness of global utility as an approximation for global CI. With extensive experiments on 29 datasets across 13 tabular generators, the authors reach some interesting conclusions, e.g., despite Bayesian Network and GOGGLE being structure learning methods, they do not perform well on structural fidelity, while diffusion-based models show their effectiveness in maintaining structural fidelity.

**Dataset Code Accessibility:**

Yes

**Dataset Code Comments:**

Dataset and code are provided in an anonymous github link.

**Ethical Considerations:**

No, there are no or only very minor ethics concerns

**Final Justification:**

The major remaining concern following the rebuttal lies in the high similarity, both in terms of conceptual framing and implementation, between the proposed global utility metric and the existing MLE method.

In the rebuttal, the authors emphasized three key differences:

1.	The proposed metric is intended for single-table causal relation measurement, whereas ClavaDDPM’s MLE was designed for multi-table relations.

2.	The global utility metric aims to mitigate bias.

3.	It is claimed to better capture causal relationships between features.

The first point is not particularly convincing, as MLE was originally designed to measure inter-feature correlations in a single table, and the causal relationships targeted by the proposed metric are essentially a subset of that broader scope. The second point is supported by some intuitive explanations in the rebuttal, which partially clarify the motivation behind the design and should be incorporated into the manuscript. Regarding the third point, the authors provide empirical evidence showing that global utility has a higher correlation with global CI, which offers more direct support for its effectiveness in capturing inter-feature causality.

Overall, the rebuttal offers some valuable clarifications. Assuming these are reflected in the revised manuscript, they would help improve its clarity and positioning. Therefore, I am raising my score to a 3. However, the main proposed metric remains too similar to prior work, and the differences in implementation are relatively intuitive, limiting the novelty and overall contribution of the paper to below the threshold for acceptance.

**Limitations Weaknesses:**

1. The proposed metric, *global utility*, which treats each column as the target and predicts it using the remaining columns, is not novel. It has already been introduced and applied in prior work, such as ClavaDDPM [1] (Section C.3.3), where similar evaluations were performed on datasets lacking ground truth causal structures. That work reports average prediction results across columns (average R² for numerical columns and average F1-score for categorical columns). If the authors intend to claim this as a major novelty, there needs to be a clear comparison between their proposed utility metric and existing approaches, highlighting what has been improved or differentiated.


2. In Section 2, the authors categorize tabular data evaluation in the literature into three areas: *density estimation, privacy preservation*, and *ML efficacy*. They argue that density estimation "fails to capture inter-feature interactions", and ML efficacy does not focus on them either—leaving the impression that there is a significant gap in explicitly measuring inter-feature interactions. However, this claim overlooks existing efforts: many works and libraries in this domain, such as TabSyn [2], TabDiff [3], and SDMetrics [4], already apply metrics that explicitly capture inter-feature interactions. These include *Pearson correlation coefficients* for continuous features and *contingency similarity for categorical features*—both of which contribute to the “trend” reported in the manuscript and measure inter-column interactions. Furthermore, earlier works like PrivMRF [5] and PrivBayes [6] have directly evaluated high-dimensional joint distributions of columns (e.g., 5-way marginals), further challenging the claim that such metrics are lacking.

    Therefore, either the taxonomy in the manuscript omits an important class of evaluation metrics, or the assertion that existing approaches “fail to capture inter-feature interactions” is inaccurate. This leads to a potential overstatement of the contribution. To strengthen the manuscript, it is essential to compare the proposed structural fidelity metrics with existing inter-feature interaction measures, either theoretically or empirically, and justify why the proposed metrics offer an improvement. Intuitively, correlation-based metrics and multi-way marginals offer continuous, fine-grained assessments of feature dependencies, even in scenarios with partial correlation—common in real-world datasets. In contrast, the proposed CI score is an aggregation of binary indicators, which is a more coarse-grained view (e.g. exising metrics can handle the cases where every feature is partially correlated, while this might be a challenge for CI). It is critical to clarify how the proposed metrics offer distinct benefits compared to the existing metrics in the manuscript.

3. The authors hypothesize that global utility *approximates global CI*, and provide empirical evidence in Section 4.2 by analyzing the correlation between the two metrics. However, this formulation may not be entirely accurate. Ideally, a high-quality synthetic dataset should capture all conditional independencies (as CI requires), but the reverse is not necessarily true, where solely maintaining the independence structure among features does not guarantee high utility. This implies that global utility may be a stronger or more comprehensive metric than global CI. The wording should be revised to emphasize the observed correlation, rather than asserting that global utility approximates global CI.


4. In Section 4.3, the authors analyze why diffusion-based models (TabDDPM [7], TabSyn, and TabDiff) perform well in terms of structural fidelity, attributing this to their ability to learn permutation-invariant conditional distributions and to the independent Gaussian noise added to each feature. However, this explanation primarily applies to TabSyn. In contrast, TabDDPM and TabDiff employ separate diffusion processes to learn different types of columns: TabDDPM uses Gaussian diffusion for numerical columns and multinomial diffusion for categorical columns; TabDiff uses Gaussian diffusion for numerical columns and masked diffusion for categorical columns, therefore the denoising process for each feature is not explicitly conditioned on features of the other type, and the noise added is not necessarily Gaussian. Therefore, the unified reasoning given does not fully apply to these models.

<br>

**References:**

[1] ClavaDDPM: https://arxiv.org/abs/2405.17724

[2] TabSyn: https://arxiv.org/abs/2310.09656

[3] TabDiff: https://arxiv.org/abs/2410.20626

[4] SDMetrics: https://docs.sdv.dev/sdmetrics

[5] PrivMRF: https://www.vldb.org/pvldb/vol14/p2190-cai.pdf

[6] PrivBayes: https://dl.acm.org/doi/10.1145/3134428

[7] TabDDPM: https://arxiv.org/abs/2209.15421

**Strengths Contributions:**

1. The paper has a clear writing style and strong logic throughout, starting from identifying the missing piece in current tabular data evaluation literature—structural fidelity—to proposing global CI as an oracle method with ground truth SCM, and finally proposing global utility as a proxy and complement to global CI for real-world datasets without ground truth. The reading experience is smooth.


2. The paper conducts comprehensive experiments on most of the representative tabular data generators, with an accurate taxonomy of their types.


3. The collected datasets, especially the real-world ones, are much more challenging compared to those used in prior literature. A few of the tables contain more than 100 columns, which could potentially drive future tabular data synthesis research toward larger-scale, more practical settings.

---

> ### Author Rebuttal · Authors · 2025-07-30
>
> Thank you for the detailed review and constructive feedback! We address all of your concerns and comments below (Table R1). We will update the manuscript accordingly to include the additional clarifications and full results of new experiments.
>
> *Table R1:* Number of our responses to the reviewers’ comments (`#Raised`/`#Replied`).
>
> | Reviewer | # Limitations/Weaknesses | # New experiments |
> | --- | --- | --- |
> | `8PZA` | 4/4 | 1 |
>
> > W1: The proposed metric has been introduced in prior work, where … lacking ground truth causal structures.
> >
>
> Global utility makes a novel contribution to the field of tabular data evaluation by providing an SCM-free lens for the underlying causal structures. Below, we detail the key differences between our proposed global utility metric and the MLE metric introduced in ClavaDDPM `[1]`. A concise summary is in Table R2.
>
> **(1) Different problem scopes.**  ClavaDDPM investigates relationships across multiple tables in a relational database, whereas TabStruct focuses on the inter-feature causal relations within a single table. Therefore, ClavaDDPM does *not* explicitly establish a direct connection between their MLE and causal structures.
>
> **(2) Column bias mitigation in global utility.** As defined in Equation 4, the utility for a variable is normalised against the corresponding reference performance. Therefore, global utility quantifies *relative* predictive power, while MLE reports the *absolute* predictive performance. Due to varying predictive difficulty across columns, directly averaging absolute performance, as MLE does, can result in “easy” columns disproportionately dominating the metric, analogous to aggregating absolute scores across datasets `[2, 3]`. In contrast, the normalisation design in global utility explicitly mitigates this column-specific bias, yielding more balanced evaluations.
>
> **(3) Predictor bias mitigation in global utility.** MLE in ClavaDDPM only employs XGBoost as the downstream predictor. As shown in Appendix C.2, it introduces potential predictor-specific bias and requires ensembling many predictors with hyperparameter tuning for stable results (i.e., generator rankings). In contrast, evaluation results across 23 real-world datasets (Appendix G.3) show that global utility provides stable and generalisable evaluations with three representative predictors.
>
> **(4) Different correlation with structural fidelity.** We further evaluate the correlation between MLE and structural fidelity metrics (i.e., local and global CI) across six SCM datasets.
>
> **Experimental setup**
>
> - All settings (data splitting, etc.) are the same as Section 4.
>
> **Results**
>
> - **Important:** Table R3 shows that MLE exhibits weak correlation with both local and global CI scores. This highlights a fundamental difference: MLE lacks a clear connection to structural fidelity, while global utility serves as an empirically grounded and effective metric.
>
> **In summary**, global utility is a novel SCM-free metric for assessing global structural fidelity. It addresses limitations of existing per-column predictive metrics by introducing mitigation strategies for both column-specific and predictor-specific biases. Global utility’s empirical effectiveness is supported by extensive experiments. We will revise the manuscript to better contextualise relevant prior work (e.g., ClavaDDPM: particularly their use of per-column predictive metrics), and clarify how global utility uniquely enables structural fidelity evaluation without relying on an SCM.
>
> *Table R2:* Summarised comparisons between global utility and MLE.
>
> |  | MLE (ClavaDDPM) | Global utility (Ours) | Evidence |
> | --- | --- | --- | --- |
> | **Conceptual** |  |  |  |
> | Problem scope | Inter-table relations in a database | Inter-feature causal relations in a table |  |
> | Measurement | Absolute performance | Relative to $\mathcal{D}_{\text{ref}}$ | Equation 4 |
> | Column bias mitigation | ❌ | ✅ | Equation 4 |
> | **Technical** |  |  |  |
> | Predictor bias mitigation | ❌ | ✅ | Appendix C.2 |
> | Stable w/o tuning predictor | ❌ | ✅ | Appendix C.2 |
> | **Empirical** |  |  |  |
> | High correlation to global structural fidelity | ❌ | ✅ | Table R3 |
>
> *Table R3:* Spearman’s rank correlation on six SCM datasets (p<0.001).
>
> |  | MLE (ClavaDDPM) | Global utility (Ours) |
> | --- | --- | --- |
> | Local CI | 0.29 | **0.11** |
> | Global CI | 0.42 | **0.84** |
>
> > W2: The taxonomy omits an important class of evaluation metrics， or … “fail to capture inter-feature interactions” is inaccurate. … CI score is an aggregation of binary indicators, which is a more coarse-grained view
> >
>
> **First**, thank you for highlighting the rich set of metrics that assess inter-feature interactions in tabular data. We *do* agree that some density estimation and ML efficacy metrics can indeed capture aspects of inter-feature interactions. Our intent was *not* to dismiss these contributions, but rather to emphasise that such metrics generally do *not* explicitly evaluate whether the causal structure of the reference data is preserved. We acknowledge that the wording may cause ambiguity here. In the revised manuscript, we will modify the writing accordingly throughout the manuscript:
>
> - We will clearly distinguish between *“inter-feature interactions”* and *“inter-feature causal interactions”*, where the latter denotes a strict subset of the former.
> - Metric descriptions: “Mainstream evaluation dimensions capture certain aspects of inter-feature interaction. For instance, Trend reflects pairwise correlation. However, they rarely assess whether the underlying causal structures are preserved. This limitation can help explain why interpolation-based methods often dominate existing leaderboards for tabular data generation (Figure 1 and Table 2).”
>
> **Second**, our density estimation evaluation aligns with the taxonomy proposed in TabSyn `[4]` and TabDiff `[5]`: *Shape* and *Trend* are treated as *low-order* statistics, while $\alpha$-precision and $\beta$-recall serve as *high-order* measures of joint-distribution fidelity (Appendix B.1). We will update our wordings on “inter-feature interactions” correspondingly.
>
> **Third**, as discussed in Appendix G.4, global CI is *complementary* to, rather than interchangeable or directly comparable with, density estimation metrics. Specifically, global CI aggregates binary CI indicators to reveal inter-feature causal relations, which are not explicitly captured by correlation metrics or multi-way marginals. For instance, correlation cannot tell a v-structure (collider) from a confounded chain, whereas CI indicators can `[6]`. Our empirical results also support this distinction (Table 2, Figure 1, and Appendix G.4). Specifically, SMOTE performs well on metrics focused on low-order correlations and high-order distributional fidelity, yet fails to preserve causal structure, as evidenced by its poor performance in global CI and global utility. This demonstrates that the granularity and scope of global CI are not directly comparable to metrics that do not explicitly attend to causal structure. Overall, global CI complements existing metrics by contributing to a more comprehensive and causally informed evaluation of tabular data generators.
>
> > W3: The wording should be revised to emphasize the observed correlation, rather than asserting that global utility approximates global CI.
> >
>
> Thank you for acknowledging the scope and expressiveness of our global utility metric. We agree that it can, in fact, offer a more comprehensive assessment than global CI: achieving a high utility score would require accurately capturing both the underlying causal structure and the associated conditional distributions. In the revised manuscript, we will accordingly amend our wording to emphasise the strong empirical correlation between global CI and global utility.
>
> > W4: The unified reasoning given does not fully apply to these diffusion models.
> >
>
> We agree that diffusion models may separately model data distributions for different feature types with different diffusion processes (e.g., noise perturbation and denoising network). We touch upon these distinctions in the implementation details (Appendix E.3). For instance, in Lines 1340-1343, we describe how TabDDPM employs distinct diffusion processes for numerical and categorical features.
>
> The core of our reasoning for the competitive performance of diffusion models lies in their ability to capture “permutation-invariant conditional distributions”. It reflects a general learning principle of diffusion models. In particular, TabDDPM and TabDiff implement this principle within each feature type, and TabSyn applies it across all features. Importantly, such a learning mechanism is agnostic to the specific form of noise used during the corruption step. As suggested, we will refine our analysis and discussion of diffusion models accordingly in the revised manuscript.
>
> **References**
>
> - `[1]` Pang, Wei, et al. "Clavaddpm: Multi-relational data synthesis with cluster-guided diffusion models." NeurIPS 2024.
> - `[2]` McElfresh, Duncan, et al. "When do neural nets outperform boosted trees on tabular data?." NeurIPS 2023.
> - `[3]` Grinsztajn, Léo, et al. "Why do tree-based models still outperform deep learning on typical tabular data?." NeurIPS 2022.
> - `[4]` Zhang, Hengrui, et al. "Mixed-Type Tabular Data Synthesis with Score-based Diffusion in Latent Space." ICLR 2024.
> - `[5]` Shi, Juntong, et al. "TabDiff: a Mixed-type Diffusion Model for Tabular Data Generation." ICLR 2025.
> - `[6]` Glymour, Clark, et al. "Review of causal discovery methods based on graphical models." Frontiers in genetics 2019.

---

> > ### Comment · Reviewer_8PZA · 2025-08-03
> >
> > Thank you for the rebuttal and the additional clarifications provided. Regarding the claimed difference in scope, I agree that ClavaDDPM is primarily designed for modeling inter-table relations, whereas TabStruct focuses on intra-table inter-feature causal relations. However, the use of MLE in ClavaDDPM still appears to serve the purpose of evaluating single-table quality. The main distinction, therefore, seems to lie in TabStruct’s explicit assertion that the learned structure corresponds to causal relations, rather than a fundamentally different approach to table-level quality assessment.
> >
> > I would also appreciate a more precise and rigorous definition of the “bias” that the proposed metric aims to address. From an implementation perspective, the key difference between global utility and MLE seems to stem primarily from the normalization term as opposed to raw utility aggregation. If this is the case, it is not entirely clear what bias is being mitigated or why this constitutes a systematic issue for MLE. For instance, the rebuttal suggests that “easy” columns could dominate the metric. Yet, within a single table, an improvement of 98% → 99% on an easy column and 10% → 11% on a hard column contributes identically to MLE, suggesting no disproportionate weighting in this sense.
> >
> > On the other hand, the proposed method may itself be susceptible to a different kind of bias. For example, if a particularly difficult column improves from 1% to 2%, this relative 100% improvement could dominate the global utility score, overshadowing improvements in more meaningful regions of the table. A clearer and more formal definition of the specific “bias” being corrected and an explanation of why the proposed normalization is preferable and would strengthen the argument.
> >
> > Lastly, I appreciate the inclusion of additional experimental results in Table R3. Could the authors please elaborate on the experimental setup used to generate these numbers? In particular, how is MLE computed when tables contain both regression and classification tasks, and how are these results aggregated for comparison?

---

> > > ### Author Response · Authors · 2025-08-04
> > > **Discussion-8PZA (part 1/2)**
> > >
> > > Thanks for the clarification. We respond to all additional comments below, and we will make the necessary updates in the manuscript.
> > >
> > > > A more precise and rigorous definition of the “bias” … why this constitutes a systematic issue for MLE … The proposed method may itself be susceptible to a different kind of bias
> > > >
> > >
> > > Thanks for clarifying. We would like to respectfully clarify that we do not claim the MLE metric proposed in ClavaDDPM *“constitutes a systematic issue”*. Indeed, we acknowledge that MLE is a meaningful and appropriate metric in the context of multi-relational data synthesis. Our main point is that MLE was not explicitly designed to assess intra-table inter-feature causal relationships; as such, it may exhibit weaker correlation with causal structural fidelity metrics. Below, we expand on the importance of addressing column bias when evaluating structural fidelity.
> > >
> > > **(1) Clarifications on column bias.** In single-table settings, “column-specific bias” refers to the variations in predictive tasks across columns when each is treated as a target variable. Moreover, we mainly focus on intra-table inter-feature causal relationships (as the reviewer correctly pointed out). Therefore, we consider predicting each column associated with different tasks (e.g., binary classification, multi-class classification, regression, etc.). A change in magnitude in predictive performance can reflect different task difficulties depending on the target variable and its type `[1, 2, 3, 4]`. In other words, the absolute performance, and its variance, across different columns are *not* comparable, therefore directly aggregating absolute scores may obscure such differences across tasks `[9]`. To mitigate this, the global utility follows a standard practice `[1, 2, 9]` to use relative performance for aggregation, which could provide a relatively more unified lens for performance across different tasks.
> > >
> > > **(2) Empirical justification for balanced evaluation.** Moreover, we find that our global utility offers a well-balanced evaluation, as it correlates strongly with global CI. By design, global CI treats all conditional independence relations uniformly and does not bias towards any specific feature. If global utility were disproportionately dominated by certain features, its correlation with global CI would be substantially weakened. However, the observed strong empirical correlation suggests that global utility exhibits the strongest correlation with global CI among the considered metrics.
> > >
> > > **(3) Impact of column-specific normalisation.** To further assess the effect of normalisation, we compare global utility computed using *absolute* predictive scores versus *relative* (i.e., normalised) scores. As shown in Table R15, using unnormalised scores leads to a substantially weaker correlation with global CI. Specifically, Spearman’s $\rho$ drops from 0.84 to 0.57. This finding supports that the normalisation design plays an important role in improving the alignment with causal structures. Furthermore, global utility based on *absolute* scores fails to produce stable generator rankings across different predictor configurations. Specifically, the Top-5 generators under Full-tuned and Light-tuned settings share *only one* generator in common when computing global utility with absolute scores. We will add a figure of the ranking instability of global utility with absolute scores (similar to Figure 7 in Appendix G.3) to the revised manuscript. This further supports that the normalisation enables global utility to deliver more robust and consistent evaluations across predictor configurations.
> > >
> > > *Table R15:* Spearman’s rank correlation with global CI across six SCM datasets (p<0.001).
> > >
> > > |  | Global CI $\uparrow$ |
> > > | --- | --- |
> > > | Global utility (absolute performance) | 0.57 |
> > > | **Global utility (relative performance)** | **0.84** |

---

> > > ### Author Response · Authors · 2025-08-04
> > > **Discussion-8PZA (part 2/2)**
> > >
> > > > The experimental setup used to generate these numbers (Table R3) … How is MLE computed … how are these results aggregated for comparison?
> > > >
> > >
> > > As stated in our initial response, we evaluate the correlation between MLE and global CI with the identical experimental setups as global utility (Section 4). Below, we detail the procedure and provide references to the manuscript.
> > >
> > > - **Step-1: Data processing** (Lines 241-243 and Appendix E.2). For each SCM dataset, we perform 10 repeats of train-validation-test splits to ensure robustness of evaluation.
> > > - **Step-2: Generator training** (Lines 243-247 and Appendix E.3). We tune the generators with Optuna `[5]` to minimise their average validation loss over the 10 repeated runs.
> > > - **Step-3: Generator inference** (Lines 1289-1291 in Appendix E.3). Each trained generator produces a synthetic dataset of equal size to its reference data.
> > > - **Step-4: MLE computing**. *We strictly follow the exact MLE evaluation procedure proposed in ClavaDDPM* `[6]` *and the implementations in its official codebase* `[7]`. For each column, we treat it as the prediction target. Then, we train an XGBoost `[8]` model using the remaining columns. For numerical targets, we average their $R^2$ scores to evaluate regression performance (`MLE-R2`). For categorical targets, we average their $F_1$ scores to evaluate classification performance (`MLE-F1`).
> > > - **Step-5: Correlation computing**. We compute Spearman’s rank correlation between global CI and each of the two MLE variants (`MLE-R2` and `MLE-F1`) across six SCM datasets. In Table R3, we report the average Spearman’s $\rho$ of the two MLE variants as a general indicator due to the limited rebuttal space. In Table R16, we provide the full results, and we will include these in the revised manuscript along with a more comprehensive discussion. Table R16 shows that both `MLE-R2` and `MLE-F1` exhibit relatively weak correlations with global CI, suggesting that the MLE may be less suitable for evaluating inter-feature causal relationships in single-table settings.
> > >
> > > *Table R16:* Spearman’s rank correlation with global CI across six SCM datasets (p<0.001).
> > >
> > > |  | Global CI $\uparrow$ |
> > > | --- | --- |
> > > | `MLE-R2` | 0.40 |
> > > | `MLE-F1` | 0.44 |
> > > | **MLE (reported in Table R3)** | 0.42 |
> > >
> > > **References**
> > >
> > > - `[1]` Feurer, Matthias, et al. "Auto-sklearn 2.0: Hands-free automl via meta-learning." Journal of Machine Learning Research 23.261 (2022): 1-61.
> > > - `[2]` Wistuba, Martin, Nicolas Schilling, and Lars Schmidt-Thieme. "Learning hyperparameter optimization initializations." 2015 IEEE international conference on data science and advanced analytics (DSAA). IEEE, 2015.
> > > - `[3]` Yogatama, Dani, and Gideon Mann. "Efficient transfer learning method for automatic hyperparameter tuning." Artificial intelligence and statistics. PMLR, 2014.
> > > - `[4]` Grandini, Margherita, et al. "Metrics for multi-class classification: an overview." *stat* 1050 (2020): 13.
> > > - `[5]` Akiba, Takuya, et al. "Optuna: A next-generation hyperparameter optimization framework." Proceedings of the 25th ACM SIGKDD international conference on knowledge discovery & data mining. 2019.
> > > - `[6]` Pang, Wei, et al. "Clavaddpm: Multi-relational data synthesis with cluster-guided diffusion models." Advances in Neural Information Processing Systems 37 (2024): 83521-83547.
> > > - `[7]` ClavaDDPM official implementation, `ClavaDDPM/eval/eval_mle.py` script.
> > > - `[8]` Chen, Tianqi, and Carlos Guestrin. "Xgboost: A scalable tree boosting system." Proceedings of the 22nd acm sigkdd international conference on knowledge discovery and data mining. 2016.
> > > - `[9]` Grinsztajn, Léo, Edouard Oyallon, and Gaël Varoquaux. "Why do tree-based models still outperform deep learning on typical tabular data?." Advances in neural information processing systems 35 (2022): 507-520.

---

> > > > ### Comment · Reviewer_8PZA · 2025-08-05
> > > >
> > > > Thank you for the further clarifications, particularly regarding the correlation comparison experiments. While the implementation of the proposed global utility metric shares high similarities with existing MLE approaches, limiting the novelty and overall contribution, the additional experiments demonstrating its correlation with global CI offer more compelling empirical evidence for its effectiveness in measuring causal relationships. Based on this, I will raise my score to a 3.

---

> > > > > ### Author Response · Authors · 2025-08-05
> > > > >
> > > > > Thank you for the response. We are pleased that we were able to address your concerns in our rebuttal and discussion. We are encouraged to hear that you find our proposed metric “*effective in measuring causal relationships”*.
> > > > >
> > > > > We would like to clarify that our work provides far more benefits than existing work (see Tables R2, R3, R15, and R16) and includes more novelties and contributions (as discussed in our paper, rebuttal and highlighted by the reviewer in their “Strengths/Contributions”). These include contributions that are :
> > > > >
> > > > > - **Conceptual:** We introduce TabStruct, *“a new benchmark framework”* for tabular generators, as rightly noted by the reviewer. Within the framework, we further introduce global utility, an SCM-free metric to enable structural fidelity evaluation on real-world tabular data – a task that remains challenging for existing MLE approaches (Table R2, R3, R15, and R16).
> > > > > - **Technical:** We provide TabStruct as an open-source benchmark suite to the community, offering one of the broadest evaluation scopes to date (Table 1 and Appendix B.4). As acknowledged by the reviewer, the evaluation is “*extensive and comprehensive”*.
> > > > > - **Empirical:** We provide actionable insights into tabular data generation. The reviewer highlights that our conclusions are “*interesting”,* and “*drive future tabular data synthesis research toward larger-scale, more practical settings”.*
> > > > >
> > > > > Given these contributions and comments, we believe our work aligns well with the aims and scope of the Datasets & Benchmarks Track. Moreover, we believe that we were able to address your concerns outlined in your review and follow-up – which would justify a positive rank. If there are other outstanding concerns that would prevent that, we are keen to discuss and resolve them.

---

### Decision · Program_Chairs · 2025-09-18

**Decision:**

Reject

**Comment:**

This paper proposes TabStruct, a benchmark for evaluating synthetic tabular data generators with a focus on structural fidelity, i.e., whether generated data preserves inter-feature dependencies and causal structures. To make this evaluation feasible across real-world datasets, the authors propose a new metric, global utility, as a proxy for conditional independence–based structural fidelity. The benchmark provides large-scale experiments across 29 datasets and 13 tabular generators, highlighting trends such as the strong performance of diffusion models.

The reviewers agree that the paper addresses an important and underexplored problem, and they appreciate the comprehensive experimental setup, clarity of writing, and potential impact of the benchmark. However, one reviewer raises strong concerns regarding the novelty of the global utility metric, noting similarities to prior methods (e.g., ClavaDDPM) and questioning whether the contribution is overstated. Others highlight additional limitations, including computational cost, reliance on correlations rather than causal guarantees, and possible bias in global utility toward strong predictive features. Nevertheless, the rebuttal provided clarifications that convinced most reviewers that the work remains a meaningful and impactful step for the community, even if some novelty questions persist. With two accept scores, one borderline accept, and one borderline reject, the consensus leans positive with an average score 4.25. Overall, I recommend acceptance, as the paper introduces a timely benchmark, contributes useful empirical insights, and is likely to stimulate further research on structural evaluation of tabular data generation.

===== FINAL UPDATE FROM DB Track PCs ====

The final decision for this paper has been taken by the program chairs after consultation with the SACs. All Senior Area Chairs have ranked papers according to the feedback from the AC during the review process. We decided to leave the original meta-review to reflect the opinion of the AC in light of the initial discussions with reviewers and SAC.